# Seasonal foraging behavior of Weddell seals in relation to oceanographic environmental conditions in the Ross Sea, Antarctica

Hyunjae Chung[1, 2†], Jikang Park [3, 4], Mijin Park[1, 2], Yejin Kim[1, 5], Unyoung Chun[6], Sukyoung Yun[4], Won Sang Lee[4], Hyun A Choi[7], Ji Sung Na[4], Seung-Tae Yoon[7,8*], Won Young Lee[1,9*†]

[1]Division of Life Sciences, Korea Polar Research Institute, Incheon 21990, Republic of Korea
[2]Department of Biological Sciences, Seoul National University, Seoul 08826, Republic of Korea
[3]Department of Biology, Kyung Hee University, Seoul 02447, Republic of Korea
[4]Division of Glacier and Earth Sciences, Korea Polar Research Institute, Incheon, 21990, Republic of Korea
[5]School of Earth and Environmental Sciences, Seoul National University, Seoul 08826, Republic of Korea
[6]Division of EcoScience, Ewha Womans University, Seoul 03760, Republic of Korea
[7]School of Earth System Sciences, College of Natural Sciences, Kyungpook National University, Daegu 41566,
Republic of Korea
[8]Kyungpook Institute of Oceanography, Kyungpook National University, Daegu 41566, Republic of Korea
[9]Polar Science, University of Science and Technology, 217 Gajeong-ro, Daejeon, 34113, Republic of Korea

*Correspondence to*:
Seung-Tae Yoon, School of Earth System Sciences, College of Natural Sciences, Kyungpook National University,
Daegu 41566, Korea
Email: styoon@knu.ac.kr
Won Young Lee, Division of Life Sciences, Korea Polar Research Institute, Incheon 21990, Korea.
Email: wonyounglee@kopri.re.kr

[†]Current Address:

Division of Glacier and Earth Sciences, Korea Polar Research Institute, Incheon, 21990, Republic of Korea

**Abstract**

Understanding the foraging behavior of marine animals in Antarctica is crucial for assessing their ecological significance and responses to environmental changes, such as seasonal changes in seawater or light hours. However, studying their responses to these seasonal changes remains challenging due to the difficult logistics of conducting observations, particularly during the harsh austral winter months. In this study, we investigated the influence of changes in seawater properties and light conditions on the seasonal foraging behavior of Weddell seals (*Leptonychotes weddellii*) in the Ross Sea, Antarctica. We affixed 35 Weddell seals with CTD tags for three consecutive years from 2021 to 2023 to record their locations and dive profiles, including depth, head acceleration, temperature, and salinity. We found that seals foraged more frequently in modified shelf water and ice shelf water than in Antarctic surface water. This preference could be connected to greater food availability. Seals also dove to greater depths and displayed increased activity in capturing prey during daylight hours. This behavior may correspond to the diel vertical migration of pelagic prey in response to varying light conditions. Consequently, marine fauna were confronted with distinct seasonal changes in the Antarctic environment and adjusted their foraging behaviors to respond to them. This highlights the importance of extrinsic factors in estimating their seasonal foraging behavior.

**Keywords**
Dive behavior, foraging habit, CTD, seal-tagging, bio-logging

## 1. Introduction

Marine animals must adapt to environmental changes in the Antarctic ecosystem, such as seawater and light availability fluctuations. Extrinsic factors play a vital role in their foraging success and food availability, specifically under challenging conditions such as oceanic warming, complex bathymetry and changing sea-ice covers (Speakman et al., 2020; Harcourt et al., 2021; Arce et al., 2022). Therefore, understanding how marine animals adapt to spatial and temporal shifts in oceanographic conditions is paramount. Antarctic animals are currently experiencing rapid environmental change (Schofield et al., 2010; Doney et al., 2011). Glacier melting and the associated oceanic changes pose significant challenges for these animals (Huang et al., 2011; Ainley et al., 2015; Sahade et al., 2015; Hückstädt et al., 2020). As top and mesopredators, marine animals serve as indicators for drastic changes. For example, Adélie penguins (*Pygoscelis adeliae*) had explored a newly exposed sea after calving of ice shelf for potentially high prey availability (Park et al., 2021), and Southern elephant seals (*Mirounga leonine*) and Weddell seals (*Leptonychotes weddellii*) had been reported to shift their foraging locations and depths with the sea ice extent and oceanographic conditions (Bailleul et al., 2007; Labrousse et al., 2021).

The Ross Sea is the largest (2.09 million km$^2$) Marine Protected Area (MPA) worldwide owing to its ecological significance (Brooks et al., 2021). It also stands as the largest continental shelf region in Antarctica. The Ross Sea has been preserved as a primary habitat for predatory animals, maintaining a pristine ecosystem because of its limited human accessibility (Smith et al., 2012). Notably, 40% of Weddell seals, 38% of Adélie penguins, and 26% of emperor penguins (*Aptenodytes forsteri*) worldwide, along with a majority of South Polar skuas (*Stercorarius maccormicki*) in the Pacific sector, reside in the Ross Sea (LaRue et al., 2021; Smith et al., 2012). In the coastal polynyas of the Ross Sea, dense shelf water, a parent water mass of the Antarctic Bottom Water (AABW), is formed by strong polynyal activity (Rusciano et al., 2013; Yoon et al., 2020). This water mass contributes approximately a quarter to the total AABW production in Antarctica (Orsi et al., 1999; Orsi and Wiederwohl, 2009; Jendersie et al., 2018; Silvano et al., 2023). Hydrographic observations have been actively conducted in the Ross Sea since the 1950s, revealing changes in its marine environment due to recent climate shifts (Jacobs et al., 2002; Castagno et al., 2019; Silvano et al., 2020; Thomas et al., 2020; Yoon et al., 2020). According to these observations, hydrographic variations in the Ross Sea, including changes in the properties of shelf water, respond sensitively to air-sea interactions driven by katabatic winds and the advection of meltwater or sea ice from the Amundsen Sea (Rusciano et al., 2013; Castagno et al., 2019; Piñones et al., 2019; Silvano et al., 2020; Yoon et al., 2020). Climate-induced variations in the marine environment of the Ross Sea are anticipated to significantly impact the behavior of marine mammals. However, our understanding of their responses remains limited due to logistical and technological challenges.

Recent technological advancements employing miniaturized CTDs have enabled researchers to monitor seawater temperature and salinity (Kokubun et al., 2021; McMahon et al., 2021; Zheng et al., 2021). Deep-diving seals have mainly been used in oceanographic observation studies, with seal-tagging datasets shared among researchers, particularly within polar ocean studies (Treasure et al., 2017). In addition to physical oceanographic data, behavioral data, such as diving patterns and acceleration serve as valuable indicators for estimating underwater foraging. Detailed feeding indices can be estimated from foraging diving depths and prey capture movements (Viviant et al., 2010; Volpov et al., 2015; Heerah et al., 2019; Nachtsheim et al., 2019; Photopoulou et al., 2020; Aubone et al., 2021).

Weddell seals are resident and primarily forage within the continental shelf of the Ross Sea (Harcourt et al., 2021; Goetz et al., 2023). Within this region, their primary diet consists of fish (notothenioids), supplemented by minor dietary components such as cephalopods and invertebrates (Dearborn, 1965; Plötz et al., 1991; Burns et al., 1998; Goetz et al., 2017). They are ranked as the deepest diving phocid species except the southern (*Mirounga leonine*) and northern elephant seals (*Mirounga angustirostris*). Hence, Weddell seals have been used to collect oceanographic and behavioral data at depths exceeding 600 m (Heerah et al., 2013; Zheng et al., 2021). These seals endure energetically demanding periods during the austral spring and autumn (October–February) seasons, marked by colony formation for pup birthing, rearing pups, breeding, and molting, often leading to considerable weight loss (Wheatley et al., 2006; Harcourt et al., 2007; Wheatley et al., 2008). Although both male and female Weddell seals sporadically forage during the reproductive season, they are classified as capital breeders that rely on energy reserves accumulated before breeding (Harcourt et al., 2007;

Wheatley et al., 2008; Goetz et al., 2017). Consequently, the overwintering period (February–September) may be critical for seals to replenish their body mass and condition.

In this study, we aimed to examine the foraging behavior of Weddell seals in association with the seasonal changes during Antarctic summer to winter seasons (March to July) using acceleration-combined CTD data from seal-tagging observations in the Ross Sea. By categorizing different water masses, we examined seasonal preferences of the seals for specific water masses. In addition, we estimated foraging behavior in response to daylight conditions.


## 2. Materials and Methods
### 2.1 Study area and CTD deployment
We conducted seal-tagging in January or early February of 2021, 2022, and 2023 along the shores of Jang Bogo Station (74°37′26″ S, 164°13′44″ E) and Gondwana Station (74°38′7″ S, 164°13′18″ E) situated in Terra Nova Bay,
Ross Sea, Antarctica (Fig. 1). We approached Weddell seals on the shore to deploy 57 CTD-Satellite Relay Data Loggers (CTD-SRDLs) or 7 CTD-SRDLs with GPS (weight: 545 g, size:105 × 70 × 40 mm, SMRU, UK). Among the 57 CTD-SRDLs affixed to individuals (19, 16, and 22 in 2021, 2022, and 2023 respectively), 55 were attached to their head, and two (ID 329 and 330, approached in 2021) were secured to their backs. Additionally, seven CTD-SRDLs (five in 2021 and two in 2022) with GPS technology were attached to their backs. Among the 64 seals, 27 were
identified as females, and 35 were males based on their morphological feature. Two were not clearly distinguished in the field; hence, these were excluded from the model analysis for comparing the sexes (see Supplementary Table 1). These devices have temperature, conductivity, and pressure sensors, which collect hydrographic data. According to the specifications of the sensors of CTD-SRDL, the accuracy of temperature, pressure, and conductivity are ±0.005 ℃, 2 bBar, and ±0.01 mS cm$^{-1}$ (SMRU Instrumentation, 2024). However,
low-resolution vertical profiles used in this study have a relatively lower accuracy for temperature (±0.04 ℃) and salinity (±0.03 g kg$^{-1}$) (Siegelman et al., 2019). All data obtained from CTD-SRDLs were received via Argos satellites and no instruments were recovered. Detailed information on the tagged individuals is provided in Supplementary Table 1.

     Before deployment, we used an anesthetic (Zoletil®50, Virbac Laboratoires, Carros, France; a
combination of 125 mg tiletamine and 125 mg zolazepam in a 50 ml solution) administered through a blowpipe. Before anesthetization, the body size was roughly estimated by the field researchers. Then, an appropriate dosage of anesthetic was administered using the proportional relationship between the body length and mass of the Weddell seals (Noren et al., 2008). The dosage administered to each individual (2 to 5 ml) is included in Supplementary Table 1. We note that two individuals (ID 329 and 330) in February 2021 were approached with
a canvas bag (McMahon et al., 2000) and no anesthesia was treated. Following the injection, we waited for over 10 minutes until the seals were sufficiently sedated. Once the seals exhibited no response to the researcher's approach, we proceeded to affix a CTD device to the seal's head using Loctite glue (Loctite 401 was used in 2021; Loctite 422 in 2022 and 2023) or Araldite epoxy resin (Araldite 2012).

     Prey capture attempts were estimated from the transmitted head acceleration data obtained from the
accelerometer embedded in the CTD tags (referred to as "accelerometer processing," as detailed in the SMRU Instrumentation manual 2023). The accelerometer mounted on this tag was initially configured to measure the three-axis acceleration at 25 Hz. However, due to network bandwidth limitations, summarized information was transmitted in lieu of complete acceleration data. To summarize prey capture behavior, the total jerk (m s$^{-3}$), the time derivative of acceleration, was calculated using the method outlined by Ydesen et al. (2014). For each
second, the tag compared the maximum value of the root-mean-square (RMS) jerk to a threshold of 250 m s$^{-3}$ to ascertain the occurrence of a prey capture attempt (PrCA) within that specific second. If the RMS jerk exceeded the threshold for several consecutive seconds, it was considered a single PrCA event. Due to bandwidth limitations, summarized information was transmitted by dividing dives into three phases (descent, bottom, and ascent) and indicating the phase in which PrCA occurred, instead of transmitting the exact time and depth. Each
dive was fitted to 12 broken-stick points (i.e., the depth at the first point below the dive threshold (6 m), ten internal points, and the final point before the dive threshold (6 m)). Dive descents were defined as the start of

the dive until the first internal point that exceeded 75% of maximum dive depth. Similarly, the ascent phase began at the first internal point, where depths exceeded 75% of the maximum dive depth, and ending after the dive. The tags computed the number of PrCA events for each phase and subsequently transmitted through a satellite network system.

## 2.2 Hydrological data

### 2.2.1 Quality control for hydrographic data

Temperature and salinity profiles obtained from seal-tagging observations were quality controlled by standard procedures widely used for the low-resolution ascent profiles of instrumented seals (Supplementary Fig. 1a; Boehme et al., 2009; Roquet et al., 2011; Siegelman et al. 2019). The procedure comprises three steps: Tag-by-tag visualization, pressure effect correction, and delayed mode calibration.

In step 1, we checked reasonable ranges of temperature and salinity in Terra Nova Bay, Ross Sea, using historical (2014–2018) ship-based CTD data and ocean mooring (sourced from Yoon et al., 2020) and removed outliers from 2021 2022 and 2023 seal-tagging data (Supplementary Fig. 1a). Subsequently, we applied the density removal algorithm regarding the minimum $N^2$ ($N$ is the Brunt-Väisälä frequency) threshold as $1 \times 10^{-9} s^{-2}$. Vertical profiles of $N^2$ show that the density removal algorithm was successfully applied to the three years of seal data (Supplementary Figs. 1b, 1c, and 1d). We found 500, 1630, and 3333 irregular profiles out of the 3315, 7552, and 7654 seal-tagging profiles recorded in 2021, 2022, and 2023 respectively, through Step 1 Therefore, we used 2815, 5922, and 4321 profiles from Step 2.

In Step 2, we corrected the pressure effect for the temperature and salinity profiles using at-sea experimental data (Supplementray Fig. 2a; Roquet et al., 2011). The in situ calibration constituted a ship-based calibration cast for CTD-SRDL sensors, attaching CTD-SRDL sensors to the CTD frame of the ship. Using at-sea experimental data, we derived linear relationships between temperature and salinity differences for each CTD-SRDL sensor, and ship-based CTD data according to pressure (Roquet et al., 2011). Temperature and salinity biases were subsequently removed from the entire profile of each tag according to the pressure calculated from each relationship (Roquet et al., 2011). The calibration cast was conducted only before the 2022 deployment; therefore, Step 2 was conducted only for the 2022 seal-tagging data.

Finally, in Step 3, we implemented a delayed-mode calibration approach to correct the offsets in the temperature and salinity profiles. Here, we used the High Salinity Shelf Water (HSSW) method (Supplementary Fig. 1), as an alternative to the LCDW method generally used for correcting seal data in the Southern Ocean (Roquet et al., 2011) because LCDW is rarely found in the continental shelf region of the Ross Sea (Budillon et al., 2011). HSSW, characterized by a homogeneous layer (Yoon et al., 2020), offers a highly stable absolute reference for estimating offsets of seal-tagging data in Terra Nova Bay (TNB). Approximately one month after the 2021, 2022, and 2023 deployments, we conducted full-depth CTD casts at 56, 43, and 69 stations within Terra Nova Bay, Ross Sea, from December 6 to 25, 2020, March 15–19, 2022, and December 3–17, 2023, respectively, aboard the ice-breaking research vessel ARAON (Fig. 2 and Supplementary Fig. 2). Absolute values from ship-based CTD can be regarded as actual values because all CTD sensors of IBRV ARAON were sent to SeaBird Electronics (SBE; Manufacturer) for sensor calibration one year before the observation period. We adjusted offsets of the seal-tagging data by comparing the salinity and temperature of HSSW within the TNB observed from ship-based CTD profiles with those from seal-tagging profiles. Potential density over 28 kg m$^{-3}$ and potential temperature below -1.9 ℃ were used as criteria for HSSW (Yoon et al., 2020).

The salinity offset range for the 2021 seal data was from -0.16 to -0.03, and the temperature was not adjusted because the temperature of HSSW from the 2021 seal data was consistent with those from the ship-based CTD data. Temperature and salinity offsets for 2022 seal data were estimated as -0.03–0.23 ℃, and -0.38–-0.01, respectively. Temperature and salinity offsets for 2023 seal data were estimated as -0.01–0.27 ℃, and -0.41–-0.01, respectively. As depicted in Figure 2 and Supplementary Fig. 2, QC completed seal data show consistent features from ship-based CTD data in Terra Nova Bay, Ross Sea.

Furthermore, we classified these water masses based on potential temperature and potential density to investigate the spatial and temporal variations in water masses within the continental shelf region of the Ross

Sea, we classified these water masses based on potential temperature and potential density (Yoon et al., 2020). Potential temperature and potential density criteria for HSSW are defined as below −1.9 ℃, and over 28 kg m$^{-3}$, respectively. Potential temperature and potential density for ice shelf water (ISW) are defined as below −1.9 ℃, and below 28 kg m$^{-3}$, respectively. Modified shelf water (MSW) is defined as colder (warmer) than −0.5 ℃ (−1.9 ℃), and denser than 27.74 kg m$^{-3}$. For modified circumpolar deep water (MCDW), the potential temperature is over −0.5 ℃, and the potential density ranges between 27.74–27.88 kg m$^{-3}$. Antarctic surface water (AASW) is defined by temperatures colder than −0.5 ℃ and densities lighter than 27.74 kg m$^{-3}$ (Fig. 2a).

### 2.2.2 Kriging

A total of 13,058 profiles were observed (2815, 5922, and 4321 in 2021, 2022, and 2023 respectively), filtered through quality control procedures. To investigate the relationship between foraging behavior and the oceanographic environment, we calculated the physical characteristics of the water column at the maximum depth of each dive. We employed the Kriging method, a commonly used technique for interpolating autocorrelated data, to calculate salinity and temperature because the oceanographic and behavioral data obtained from the CTDs did not temporally match (Oliver and Webster, 1990). Kriging was performed using the *gstat* package (Pebesma, 2004) in R, and the salinity and temperature at the maximum depth of each dive were obtained by calculating the two-dimensional space of depth and time. Water masses were classified based on these values. To account for the spatiotemporal anisotropy, we scaled the values between 0 and 1 based on the maximum and minimum values, and multiplied the time values by 50. Separate Kriging processes were conducted for the 2021, 2022, and 2023 datasets, and the reliability of the results was confirmed via five-fold cross-validation. The mean, root mean squared, and mean absolute errors for the kriging estimates are summarized in Supplementary Table 2. To create the Hovmöller diagram (Fig. 3 a and b), salinity and temperature from 1 to 600 m depth between February 15 and July 15 in 2022 were also calculated using kriging with seal-tagging profiles around Terra Nova Bay within the longitude range between 160 and 170˚E and latitude range between 76 and 74˚S.

### 2.4 Dive data classification and filtration

We distinguished between benthic and pelagic seal dives. The bathymetric depth corresponding to each dive location was assigned using bathymetry data from IBCSO (IBCSO.org, Dorschel et al., 2022). Dives characterized by a submergence depth of 80% or more of the assigned depth were classified as benthic dives (Kokubun et al., 2021). The Python package *pvlib* (Holmgren et al., 2018) determined the solar altitude at each dive location and time, with altitudes above 0 categorized as daytime and below 0 as nighttime. Dives with bathymetric values greater than 0 were excluded to eliminate inaccurately recorded dives. When seal diving was deeper than the bathymetric values, the dives were regarded as benthic dives. Furthermore, dives with durations that were too short or long and depths that were too great (dive duration = 0 s, dive duration > 5760 s, dive depth > 906 m; Heerah et al., 2013), and those characterized by vertical travel speeds exceeding 5.1 m s$^{-1}$ were excluded (Davis et al., 2003).

### 2.5 Statistics

To investigate the factors influencing the feeding behavior of Weddell seals, we set the response variable as log-transformed prey capture attempts (log(PCA_BTM + 1)) and used dive type (benthic or pelagic), season (month), sex, water mass, and year as explanatory variables to determine the minimal model through backward elimination. For the analysis, we excluded the data obtained from seals with CTD devices attached to their backs to catch the head movements. First, we compared the full model containing all explanatory variables against the models with each variable systematically removed using a likelihood test; through this process, we eliminated variables deemed non-contributory. After repeating this process, we obtained a parsimonious model containing only the important variables. Additionally, we compared all possible models created using different combinations of explanatory variables by comparing their Akaike Information Criterion (AIC) and Bayesian Information Criterion (BIC) values. We subsequently obtained the best model with the smallest AIC and BIC values

(Supplementary Tables 3 and 4). The explanatory variables of the best model obtained using the three methods (backward elimination, AIC, and BIC) were consistent. After finding the minimal model, we conducted post-hoc tests using the *multcomp* R package (Hothorn et al., 2008) to investigate differences in the categorical variables included in the minimal model (season and water mass). Additionally, after confirming the seasonal change in PrCA, we aimed to investigate whether the seasonal change differed by dive type. To do this, we included an

interaction term between Julian date (day of the year) and dive type, with sex, water mass, and year as candidate explanatory variables, and identified the best model. To examine diurnal patterns, we subsequently examined the effect of time periods on the dive depth, number of dives, and prey capture attempts. Throughout this process, we created a linear mixed-effects model using the *nlme* R package (Pinheiro et al., 2022), in which we set each identity as a random effect and included a temporal autocorrelation term. The models were estimated

using restricted maximum likelihood. To ensure the robustness of our models, we performed Monte Carlo Cross Validation (CV) with a 4:1 train-test split and 100 iterations for each model. This approach allowed us to assess the stability and generalizability of the models. The standard deviations of the R-squared values were all below 0.02, further confirming the consistency and reliability of our models.

**3. Results**
The telemetry data revealed that the Weddell seals in this study dispersed from the tagged region (near Jang Bogo Station; 62.2° S, 58.8° W) and traveled throughout continental shelf regions in the Ross Sea (Fig. 1). Among the 64,014 dives observed, 11,741 were categorized as benthic dives, while 52,273 were pelagic.

Seal CTD sensors have been used to observe five water masses in the continental shelf region of the

Ross Sea: AASW, MCDW, MSW, ISW, and HSSW (Fig. 2; Orsi and Wiederwohl, 2009). When compared to the ship-based CTD data collected in the TNB during the austral summer of the same year, the seal tagging data showed a wider range of temperature and salinity of AASW (Fig. 2). The wide range of temperature and salinity values of the AASW represents its seasonal variation, being icy cold and fresh during the sea ice melting period (mainly austral summer), and subsequently transitioning to being warm and saline due to latent heat release and brine

rejection during the sea ice formation period (mainly austral winter). The 27.8 kg m$^{-3}$ isopycnal exhibited a shoaling trend from mid-March onwards, eventually disappearing by salinity increase in the surface due to brine rejection and vigorous mixing through the whole water column by May (Fig. 3). After May, HSSW and MSW, which are colder than -1.7 ℃ and denser than 27.8 kg m$^{-3}$, were mainly identified in the TNB (Fig. 3). These results support the notion that our seal-tagging data captured the increase in the density of AASW over the period

between austral summer and winter. The dive depth shows an increasing trend from March to July as the water temperature decreased while salinity and density increased (Fig. 3c).

Moreover, the presence of MCDW was more discernable in the seal-tagging profiles compared to the ship-based CTD data obtained from the TNB, despite its limited occurrence (only 125 depths of 13,058 profiles) (Figs. 1 and 2; Supplementary Fig. 3). This prominence arises because of seals diving into the Drygalski and Joides

troughs near the continental shelf break region (Fig. 1). Among seal data, more profiles were obtained near the shelf break and the eastern part of continental shelf regions in 2021 and 2022 than those in 2023 (Fig. 1). Due to this difference in spatial sampling, MCDW was identified more clearly in 2021 and 2022 compared to 2023 (Fig. 2; Supplementary Fig. 3). Furthermore, the ISW observed across the continental shelf region of the Ross Sea demonstrates a wider salinity range than the ISW observed in the TNB (Fig. 2), consistent with previous studies

(ex. Budillon et al., 2011). In 2021, 2022, and 2023, properties of HSSW were well detected (Fig. 2) and it was mainly observed in the western part of the continental shelf region of the Ross Sea where polynyas exist (Fig. 2; Supplementary Fig. 3).

In all three years, the Weddell seals tagged in this study exhibited distinct diving behaviors across months. Figure 3 illustrates the seasonal changes in dive depth. The dive depth shows an increasing trend from

March to July, whereas the number of PrCA events decreases in June and July compared to March and April. When considering diving depth (p < 0.001; log likelihood ratio test between the best model and a model excluding the variable "season"), the shallowest dives were undertaken in April, whereas the deepest diving occurred in July (200 ± 137 m in April, 265 ± 154 m in July; mean ± standard deviation) (Fig. 3; Tables 1 and 2). In

terms of PrCA (p < 0.001; log likelihood ratio test between the best model and a model excluding the variable "season"), the highest number was observed in April, whereas the lowest occurred in June (3.29 ± 6.11 in April, 1.56 ± 2.59 in June) (Fig. 4a; Tables 3 and 4). Additionally, PrCA values varied based on water mass and dive type (benthic or pelagic) (p < 0.001 for both; log likelihood ratio test between the best model and a model excluding the variables "water mass" and "dive type"). Based on our water mass definition, Weddell Seals performed many dives (76.76% of total dives) and high frequent observations of PrCAs (86.7% of total PrCA events) in MSW. The kernel density plots of dive distributions on a TS diagram are shown in Supplementary Fig. 4. Notably, Weddell seals displayed a higher number of PrCA events per dive in HSSW, MSW, and ISW compared to AASW (additional 1.14, 0.66, and 0.65 in PrCA per dive for HSSW, MSW and ISW, respectively; Tables 3 and 5). Our seals had 0.58 more PrCA during benthic dives than during pelagic dives (Fig. 4b; Table 2), despite the fact that benthic dives were not predominant (11,741 out of a total of 64,014 dives, Fig. 4c). From March to July, PrCA consistently decreased during pelagic dives, whereas no significant decrease was observed during benthic dives (Figure 5, Supplementary Table 5).

Weddell seals demonstrated different diving behaviors between daytime and nighttime, delineated by solar altitude. During daylight hours, seals dived an average of 76.4 m deeper and had a higher proportion of benthic dives compared to nighttime (Figs. 6a and 6d, Table 6). Additionally, seals demonstrated higher PrCA events during the daytime, with an average of 4.89 foraging attempts per dive, compared to 2.13 attempts during the nighttime (Fig. 6b; Table 6). Interestingly, no discernible difference was observed in the number of dives between the day and night (Fig. 6c; Table 6).

## 4. Discussion

In this study, we observed a distinct seasonal pattern and water mass preference in the foraging behavior of Weddell seals. Shallow and deeper diving was observed in April and July, respectively, and foraging frequencies were the highest in April and lowest in June. The detected water masses from the seal-CTD were MCDW, MSW, ISW, AASW, and HSSW (. 2; Supplementary Fig. 3). Among these, Weddell seals exhibited significantly higher number of PrCA events per dive for HSSW, MSW and ISW over AASW. In contrast, MCDW was rarely detected. Furthermore, more PrCA events were observed during benthic dives than those in pelagic dives. Finally, a diel diving pattern among the seals was observed, with an increase in the proportion of benthic dives, foraging frequency, diving depths, and the number of dives during the day compared to night.

To best our knowledge, this is the first to measure the prey capture attempts of Weddell seals in winter season directly using head acceleration with CTD. Previous studies have estimated foraging behaviors from indirect information, including horizontal location, vertical swim speed, dive time, and dive depth rather than being directly measured (Nachtsheim et al., 2019; Kokubun et al., 2021; Goetz et al., 2023). While these proxies are indirect indices and should be interpreted cautiously, acceleration data like the data our CTD obtained is particularly beneficial as it can directly detect PrCA, providing a more accurate measure of foraging activity (Heerah et al., 2019; Allegue et al., 2023). This allows us to correlate foraging activities with the recorded environmental conditions, providing a clearer understanding of how these animals interact with their habitats. We presume that the combination of CTD and acceleration data offers a comprehensive view of both the physical environment and the behavioral responses of the seals, leading to more accurate and insightful conclusions.

Our results conclusively illustrate a seasonal shift in diving depth and the number of PrCAs per dive. This phenomenon can be attributed to fluctuations in oceanographic and light conditions. Notably, Weddell seals preferred MSW or ISW over AASW during their foraging dives. As the lower boundary of the AASW shifted downward during June and July, the seals engaged in progressively deeper dives during the winter months possibly to follow the MSW or ISW. Secondly, a seasonal decrease in sunlight could limit prey accessibility, particularly pelagic fish species. The number of daylight hours in this region significantly decreased from March to July. On March 1, daylight duration is over 16 h with a meridian altitude of over 23° (based on data at Jang Bogo Station); but the onset of the Polar night in early May (5th May in 2021; 6th May in 2022; 6th May in 2023) resulted in continuous darkness without sunrise. In the Ross Sea, the euphotic zone, where sufficient light for photosynthesis is available, is situated at a depth of 34 ± 13 m in spring, 26 ± 9 m in summer (mean ± standard

deviation), and within a range of 14–66 m (range) in winter (Fabiano et al., 1993; Smith et al., 2013). Below the euphotic zone lies the dysphotic zone, where light is present; however, it is insufficient for photosynthesis to occur. Based on the findings of Sipler and Connelly (2015), the dysphotic zone in the Ross Sea extends to a depth of 170 m. Notably, Antarctic silverfish and holopelagic prey in the Ross Sea are found at depths of 0–700 m (De Witt et al., 1990), and their prey abundance is high in the upper water layers (50–200 m, Mintenbeck, 2008). This implies that Antarctic silverfish may inhabit the euphotic and/or dysphotic zones. Weddell seals have been reported to use light and other senses, including vibrissal sensations, for swimming, detecting, and catching prey (Wartzok et al., 1992; Davis et al., 2004). Therefore, when sunlight is available, Weddell seals employ a combination of visual and other sensory inputs to capture pelagic or cryopelagic prey. Conversely, when sunlight is unavailable, or benthic prey are target, they must rely solely on non-visual sensory inputs for effective foraging. The diminished light conditions experienced in June and July posed challenges for seals in locating prey, thereby leading to a decrease in PrCA events per dive and an increase in diving depths during these months compared to March. Our data also showed that during the polar night in June and July, the PrCA per dive decreased in pelagic dives, while benthic dives showed no notable change (Figure 5, Supplementary Table 5). This suggests that benthic dives may play a crucial role in Weddell seals' foraging strategy during the winter months, when light conditions are diminished, making benthic prey potentially more reliable than pelagic prey.

The seasonal changes in diving behavior likely reflect corresponding seasonal changes in the distribution or composition of prey. Previous studies analyzing the diet of Weddell seals in the Ross Sea through scat or stomach contents have highlighted Antarctic silverfish as the primary pelagic prey consumed by Weddell seals across all seasons (Dearborn et al., 1965; Plötz et al., 1991; Burns et al., 1998; Goetz et al., 2017). Therefore, the increased dive depth of Weddell seals may suggest that the distribution of Antarctic silverfish, their main prey, and only holopelagic fish in the Ross Sea shifts deeper as winter approaches. Although, the seasonal variations in the vertical distribution of Antarctic silverfish remain unknown, Antarctic krill (*Euphausia superba*), one of their primary food sources, may migrate to deeper waters during winter when the sea surface is covered with ice and food in the upper waters becomes scarce (Smidt et al., 2011; Meyer et al., 2017). This could imply that Antarctic silverfish may migrate to deeper waters as winter approaches. As Antarctic silverfish mature, they tend to inhabit deeper waters (La Mesa and Eastman, 2012), suggesting a shift in the prey composition towards larger and deeper-dwelling adult Antarctic silverfish as winter approaches. Another plausible factor behind this seasonal shift in diving behavior could be a corresponding shift in dietary preferences, involving greater consumption of benthic fish than pelagic or cryopelagic fish. Additionally, seasonal variations in interspecific competition, particularly involving emperor penguins, another apex predator species in the Ross Sea year-round (Burns and Kooyman, 2001; Smith et al., 2012), could affect the foraging behavior of Weddell seals. In winter, emperor penguins must actively seek sustenance to nurture their offspring, potentially intensifying interspecific competition with Weddell seals (Burns and Kooyman, 2001). Given that the diving capacity of emperor penguins is lower than that of adult Weddell seals (Kooyman et al., 1980; Kooyman and Kooyman, 1995; Burns, 1999), Weddell seals may forage at greater depths to minimize interspecific competition. Notably, in our data, deep dives (exceeding 350 m) occurred at a rate of 22.4% in July for Weddell seals, whereas emperor penguins performed deep dives at a rate of less than 10% (Burns and Kooyman, 2001). This suggests a potential seasonal adjustment in foraging strategy, although direct evidence for this behavior remains limited.

A previous study on Weddell seals in the Ross Sea showed that seasonal changes for foraging effort were observed, with dive depth and prey search effort (estimated by search effort time in a given space) increasing from summer to winter (Goetz et al., 2023). Additionally, prey search effort was also higher in benthic dives (Goetz et al., 2023). The seasonal increase of dive depth agreed with our findings, but the prey search effort showed the contrary of PrCAs, our foraging measurement. Goetz et al. (2023) observed that the prey search effort was the highest in winter of 2010 to 2012. This could be due to the different seasonal prey availability across the seasons. During winter, our study indicated that prey capture was lower while the previous study by Goetz et al. (2023) showed the prey search effort was higher. To combine the two studies, it seems that the seals had to spend more time to search for prey despite the low foraging success in winter. Still, it is difficult to compare the two studies since there is an approximate 10-year difference. The diet composition of Weddell seals exhibits considerable interannual variability in the Ross Sea area (Goetz et al., 2017). The sea ice extent and the food availability for top predators can vary annually (Ainley et al., 2020). Such variations in sea ice extent can possibly

influence plankton blooms and the seasonal prey abundance for seals between the two studies (Arrigo et al., 2004; Lorrain et al., 2009). Our measurement also has a limitation to compare the seasonal change. Prey capture attempts were estimated by 'jerk' from acceleration sensors attached on heads. Prey capture attempts do not necessarily correlate with the quality or quantity of the prey successfully obtained. For example, jerk could be overestimated when handling larger prey items, as the number of handling movements increases (a case of Australian fur seals, *Arctocephalus pusillus doriferus*, Volpov et al., 2015). These limitations of foraging proxies may account for the observed differences in their seasonal trends.

Our seal CTD data revealed a dynamic change in vertical stratification by the seasons. During early austral fall, the water columns within the Ross Sea are structured, with HSSW, MSW, and AASW from the bottom up; however, this stratification weakens as winter advances; strong mixing owing to the influence of winds coupled with active sea ice formation at the surface diminishes the stratification over the whole water column (Figs. 3b and 3c). Additionally, ISW exists near the ice shelves instead of spreading out to the central part of the continental shelf region (Supplementary Fig. 3). This behavior might be associated with the relatively lower rates of basal melt and meltwater flux of ice shelves in the Ross Sea (Rignot et al., 2013; Rignot et al., 2019).

Weddell seals exhibited more frequent feeding behavior in HSSW, MSW, and ISW than in AASW, and they rarely ventured into MCDW. These findings might reflect the inherent nutrient composition of each water mass. HSSW is the densest water mass (potential density > 28 kg m$^{-3}$) in the Ross Sea (Budillon et al., 2011; Yoon et al., 2020), and the biological products from the surface are being transported to depth, accumulating nutrient contents of HSSW (Arrigo et al., 2008; DeJong et al., 2017; Ingrosso et al., 2022). It provides unique habitats for benthic fish for survival and distribution (La Mesa et al., 2004). In our results, dives in HSSW were mostly performed in the benthic (69.4%). Thus, HSSW may contain prey, usually benthic species, for seals.

MSW is formed by mixing shelf water with surrounding water masses, including MCDW, within the continental shelf region. MCDW is crucial in heat and nutrient cycling in the Southern Ocean because it is warm and nutrient-rich (Smith et al., 2012; Kutska et al., 2015; Gerringa et al., 2020). MCDW contains a significantly higher concentration of macro-nutrients and contributes to the basal melt of ice shelves, which may increase primary production. However, the dissolved oxygen (DO) levels in MCDW are notably low, falling below 5 mL L$^{-1}$ (Jenkins et al., 2018; Yoon et al., 2020). This is lower than the critical threshold of oxygen concentration for krill, implying that the prey availability for seals in MCDW would be limited (Brierley and Cox, 2010). However, MSW may contain high amounts of nutrients from MCDW and sufficient oxygen contents (Orsi and Wiederwohl, 2009; Smith et al., 2014). According to ship-based CTD observations with SBE43 DO sensor values in TNB during the austral summer 2021, 2022, and 2023, it was found that the DO of MSW was over 6.5 mL L$^{-1}$.

ISW is a water mass formed by melting ice shelves and is characterized by a potential temperature below the freezing point. This water mass can harbor essential nutrients, such as iron, which may be present on ice shelves; thus, potentially making ISW a nutrient source (Sedwick and DiTullio, 1997; Smith et al., 2014). Nutrient-rich hydrographic conditions may be related to the high prey availability. Weddell seals exhibit increased foraging behavior under nutrient-rich conditions in other regions (Heerah et al., 2013; Nachtsheim et al., 2019; Kokubun et al., 2021). Moreover, ISW also has relatively higher oxygen, for example, DO sensor values of ISW in TNB during the austral summer of 2021 and 2022 are higher than 6.4 mL L$^{-1}$. AASW is generally deficient in nutrients by vigorous biological processes despite the high DO. Therefore, Weddell seals could have higher PrCA events in MSW and ISW than AASW and MCDW because they satisfy rich nutrients and high-DO characteristics.

Although only 18.8% of all dives were categorized as benthic dives, more foraging attempts were observed during these dives. From an energy-efficiency perspective, the costs associated with the diving behavior of Weddell seals increase as the duration of their dives increases. In particular, dives lasting longer than 23 min entail additional anaerobic costs. Despite the substantial energetic costs associated with prolonged dives, the benthic zone is a habitat for numerous sizeable prey species weighing over 1 kg, including the Antarctic toothfish and icefish (La Mesa, 2004; Goetz et al., 2017). Hence, Weddell seals can reap substantial benefits in the benthic zone. This dynamic could result in a higher frequency of foraging attempts per dive during benthic dives than pelagic dives for larger prey. Furthermore, PrCA instances were estimated by tallying the occurrences of jerks (the temporal derivatives of acceleration) surpassing the predefined threshold (250 m s$^{-3}$), as recorded on the bio-logger attached to the head of the Weddell seal. Benthic prey in the Ross Sea predominantly comprises hefty fish, such as Icefish or Antarctic toothfish, and other fish heavier than the Antarctic silverfish,

the only holopelagic fish in the Ross Sea whose adult form exceeds 50g. Diving predators require increased mobility to effectively handle larger prey, resulting in higher variance in behavioral data, including acceleration (Watanabe and Takahashi, 2013; Volpov et al., 2015). Weddell seals also handle large prey such as Antarctic toothfish, the flesh part of which is exclusively consumed by them (Davis et al., 2004; Ainley and Sniff, 2009; Goetz et al., 2017). Therefore, this study acknowledges the likelihood that foraging frequency may have been

overestimated when Weddell seals handle larger prey.

        Differences in diving behavior between day and night were also observed. Weddell seals performed deeper dives during the day, which was marked by a higher incidence of benthic dives and PrCA events; however, no significant differences between the time periods were observed. The variation in diel patterns of diving depth could potentially be attributed to the vertical migration behavior of pelagic prey. This migration phenomenon is

well-documented among pelagic fish species, including Antarctic pelagic fish, which exhibit a diel vertical migration pattern. This fish dive to greater depths as the amount of light at the surface increases, effectively reducing their vulnerability to visual predators that rely on light to locate and pursue prey (Childress, 1995; Fuiman et al., 2002; Hays, 2003; Robison, 2003; Sutton, 2013). Moreover, seals rely on their visual senses to detect prey (Davis et al., 1999). Thus, these seals can dive to greater depths during the day, corresponding to the

migratory behavior of pelagic prey in the Ross Sea. Additionally, the energy expenditure associated with hunting pelagic prey may increase with deeper dives during the day. In contrast, the cost of hunting benthic prey may decrease as light increases. Therefore, the proportion of benthic dives increases during the day. As visual predators, Weddell seals are more adept at hunting during daytime, predominantly on their vision. Consequently, the number of PrCA events increased during the daylight hours.


**5. Conclusion**

Concurrently analyzing hydrographic and behavioral data from the Ross Sea revealed seasonal variations in the foraging behavior of Weddell seals, which was closely linked to shifts in oceanographic environmental conditions (Fig. 7; Supplementary Fig. 5). The seals demonstrated a preference for water masses, which could potentially

be both nutrient-rich and high-DO, and exhibited distinct foraging strategies depending on the light conditions during the day and night. This study demonstrates that Weddell seals adjust their foraging behavior, spatially and temporally adapting environmental factors. Over the last several decades, the hydrography of the Ross Sea has undergone considerable changes with an increasingly warming world (Castagno et al., 2019; Silvano et al., 2020; Thomas et al., 2020; Yoon et al., 2020). This suggests a continuous adaptation process in the foraging behaviors

of marine mammals, including Weddell seals, as they navigate changing marine environments. Therefore, continuous monitoring of the foraging behavior of marine mammals in the Ross Sea is necessary. Our findings serve as a baseline and establish a foundational understanding for future research, particularly concerning the impact of marine environmental changes on the ecosystem of the Ross Sea MPA.

**Data Availability**

The behavioral and oceanographic data related to this study can be accessed at the Korea Polar Data Center (KPDC) website, kpdc.kopri.re.kr. The datasets are available under the following DOIs:

- https://dx.doi.org/doi:10.22663/KOPRI-KPDC-00002402.1

- https://dx.doi.org/doi:10.22663/KOPRI-KPDC-00002401.1

- https://dx.doi.org/doi:10.22663/KOPRI-KPDC-00002077.1

- https://dx.doi.org/doi:10.22663/KOPRI-KPDC-00001658.4

**Author Contributions**

HC, S-TY and WYL designed the research and all authors contributed to the conceptualization. JP conducted the

investigation in 2020. MP, YK and UC conducted the investigation in 2021. WY and JSN conducted the investigation in 2023. HAC and S-TY performed the quality control of raw data for formal analysis. SY and WSL participated the methodology and project administration. HC did the data curation and contribute to formal analysis. HC, S-TY, and WYL prepared the original draft manuscript and revised it with comments from all authors.

**Competing Interests**

The authors declare no competing interests.

**Acknowledgments**


This work was supported by a Korea Polar Research Institute (KOPRI) grant funded by the Ministry of Oceans and
Fisheries (KOPRI PE21140, 22140, PE23140; WYL, JP, HC, MJP, YK, and UC) and supported by the Korea Institute
of Marine Science & Technology Promotion (KIMST) funded by the Ministry of Oceans and Fisheries (RS-2023-
00256677; PM23020; WYL, HC, JP, SY, WSL, HAC, JSN, and S-TY). This research was also supported by Basic Science
Research Program through National Research Foundation of Korea (NRF) funded by the Ministry of Education
(2022R1I1A3063629; S-TY, and HAC).

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

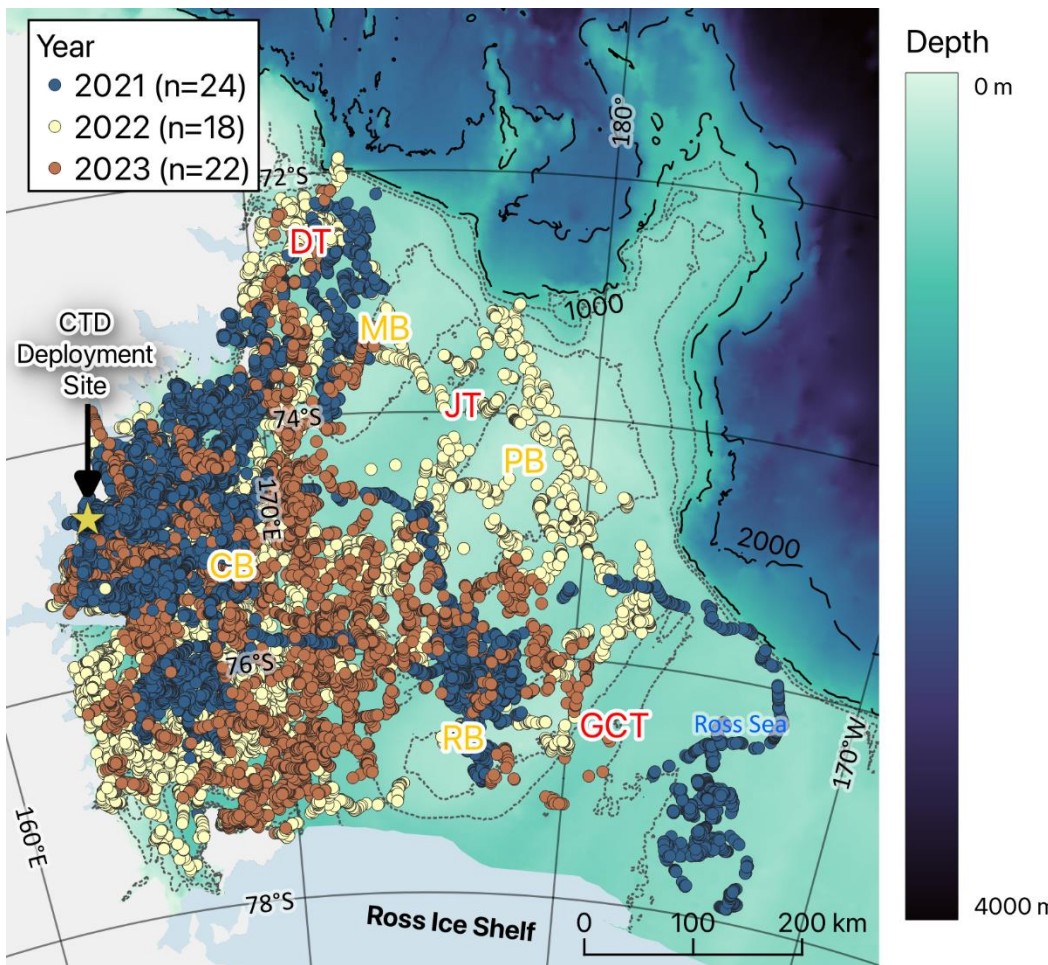

**Figure 1. Dive locations of seals tagged at Terra Nova Bay in the Ross Sea (blue, yellow, and brown dots indicated seal ARGOS locations in 2021, 2022, and 2023, respectively).** The abbreviations CB, MB, PB, RB, DB, DT, JT, and GCT mean Crary Bank, Mawson Bank, Pannell Bank, Ross Bank, Drygalski Trough, Joides Trough, Glomar Challenger Trough, respectively. The dashed line represents the shelf break (at depths of 1000 and 2000 m), while the dotted line represents bathymetry at 200 m intervals (200-800 m).

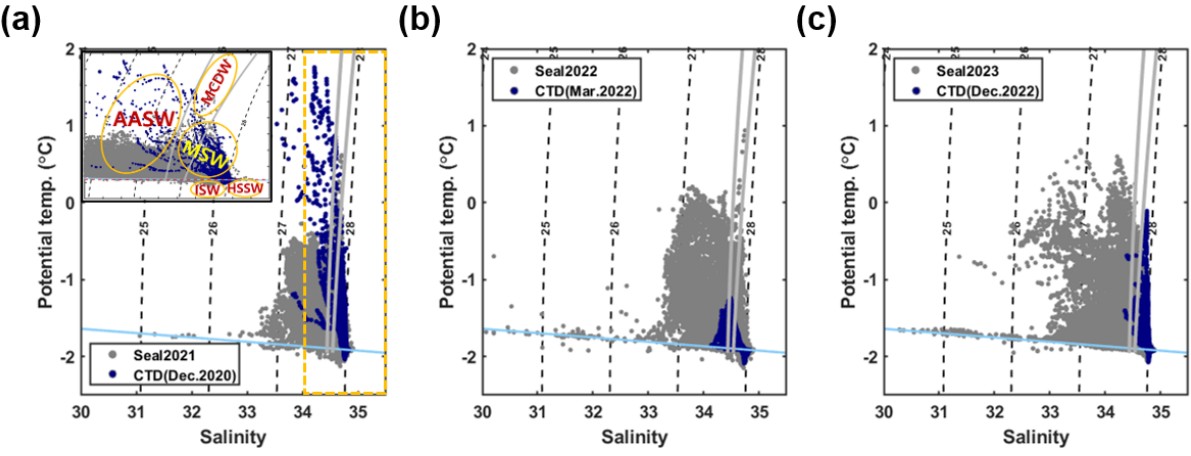

**Figure 2. ϑ–S diagram from seal tagging data and ship-based data** (a) ϑ–S diagram for seal tagging data obtained during 2021 (gray) and ship-based CTD data recorded from 6 to 25 December, 2020 (blue). The dashed black lines indicate isopycnals (kg m⁻³); solid gray lines represent 28 and 28.27 kg m⁻³ neutral density surfaces. The solid sky-blue line indicates the surface

freezing point depending on the salinity. The inset indicates a zoomed-in plot for the $\vartheta$–$S$ diagram (yellow-green dashed box) and shows the approximate temperature and salinity range of each water mass. The abbreviations AASW, MCDW, MSW, ISW, and HSSW correspond to Antarctic surface water, modified circumpolar deep water, ice shelf water, and high salinity shelf water, respectively; (b) $\vartheta$–$S$ diagram for seal tagging data obtained during 2022 and ship-based CTD data recorded from 15 to 19 March 2022, and (c) for seal tagging data obtained during 2023 and ship-based CTD data recorded from 3 to 17 December 2022.


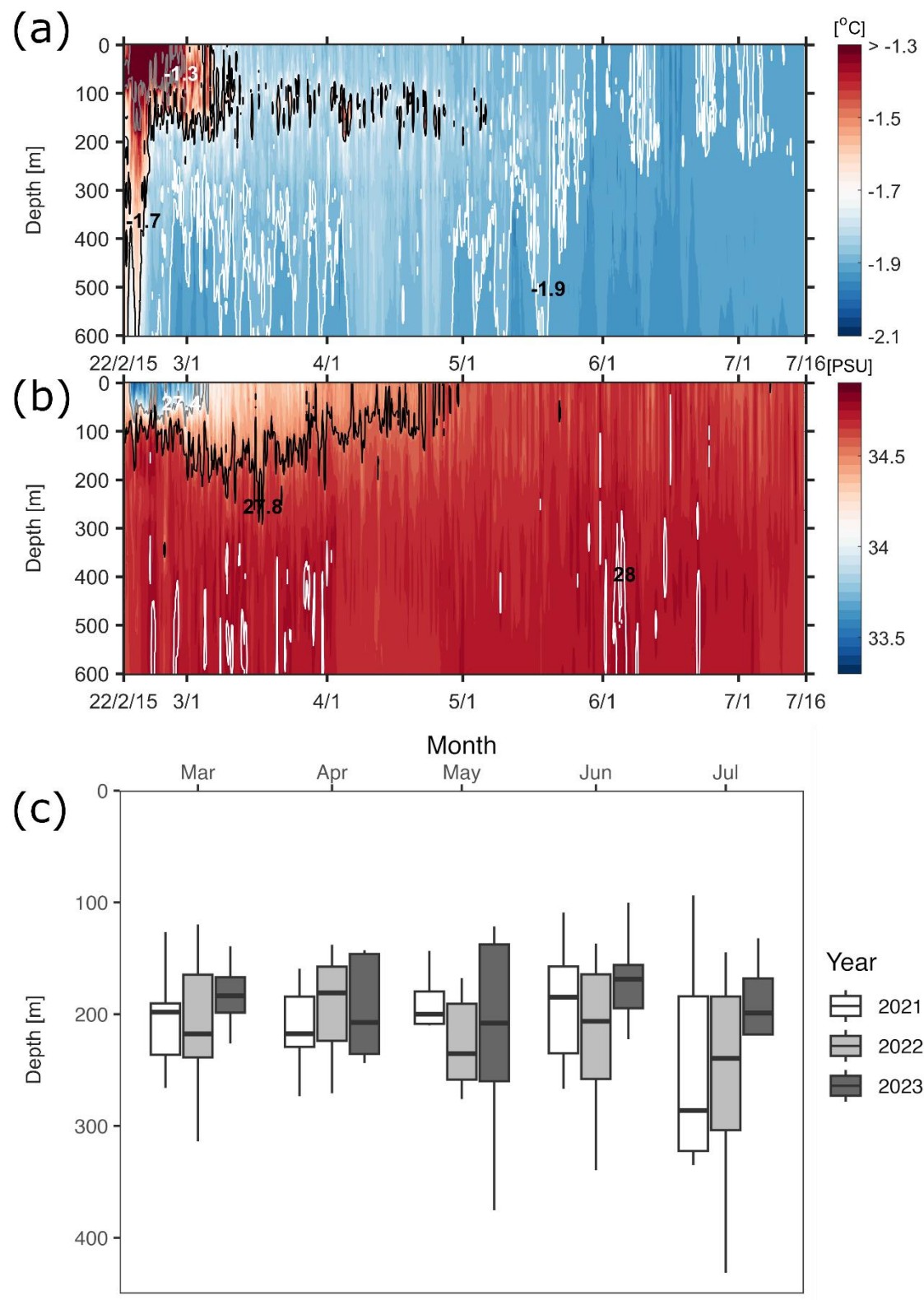

**Figure 3. Temporal variation in dive depths of Weddell seals for 2021, 2022 and 2023 with Hovmöller diagram of seawater properties around Terra Nova Bay in 2022.** (a) Hovmöller diagram of potential temperature around Terra Nova Bay. Gray,

black, and white solid lines represent -1.3, -1.7, and -1.9 ℃ isotherms, respectively. (b) Hovmöller diagram of salinity around Terra Nova Bay. Gray, black, and white solid lines represent 27.4, 27.8, and 28 kg m$^{-3}$ isopycnals ($\sigma_\theta$), respectively. (c) White, grey, and black boxes indicate diving behaviors in 2021, 2022, and 2023, respectively, showing a tendency for deeper dives as austral winter approaches.

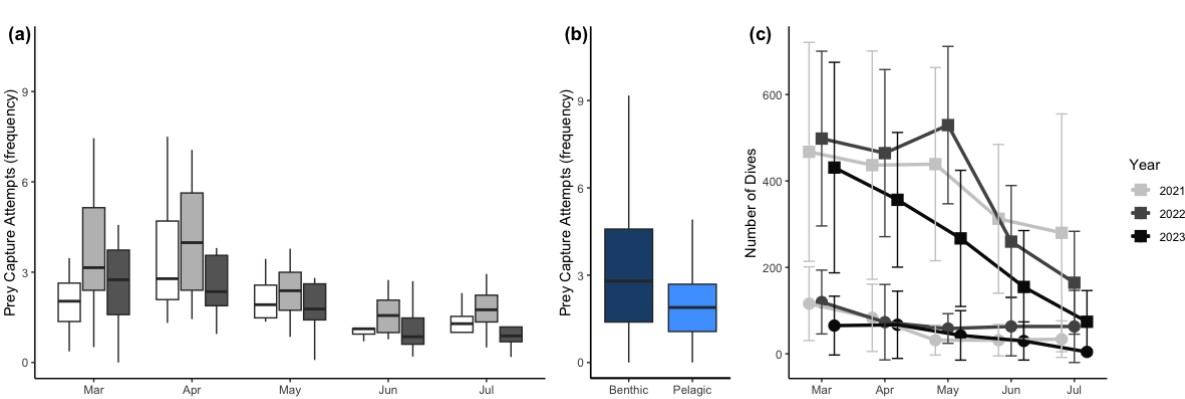

**Figure 4. Prey capture attempts (PrCA) among (a) seasons (month) and (b) dive types (benthic or pelagic) and seasonal change of the dive frequency.** Prey capture attempts were highest in April and lowest in June. Prey capture attempts were higher in benthic dives compared to pelagic dives. The dark blue box indicates the number of PrCA events per dive during benthic dives, whereas the lighter blue box represents the same statistic during pelagic dives. In (c), curves with square markers represent the total dives and curves with circle markers represent the number of benthic dives of each month. The error bars represent the mean ± standard deviation.

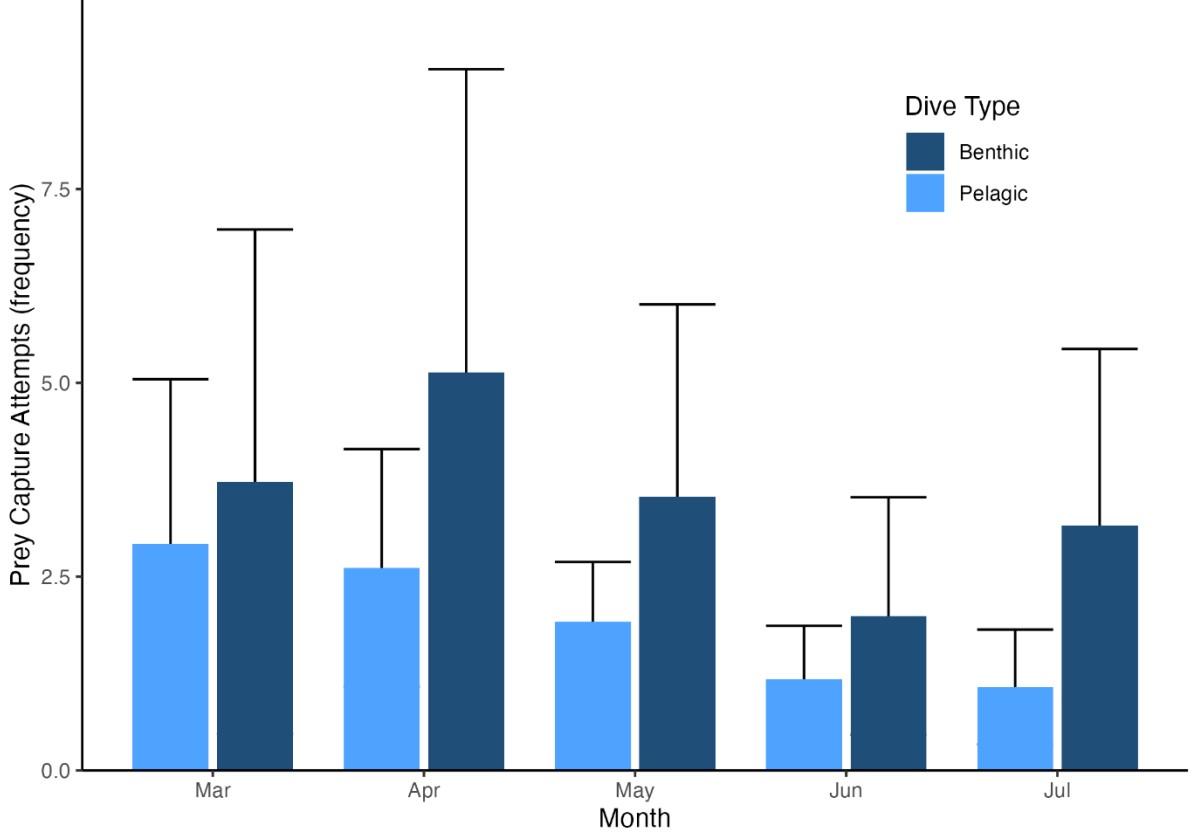

**Figure 5. Seasonal change of prey capture attempts (PrCA) per dive by dive type (benthic or pelagic dive)** PrCA were consistently higher during benthic dives compared to pelagic dives. While PrCA during pelagic dives decreased as winter approached, it decreased less significantly during benthic dives. PrCA during benthic dives is shown in dark blue, while PrCA during pelagic dives is represented in light blue.


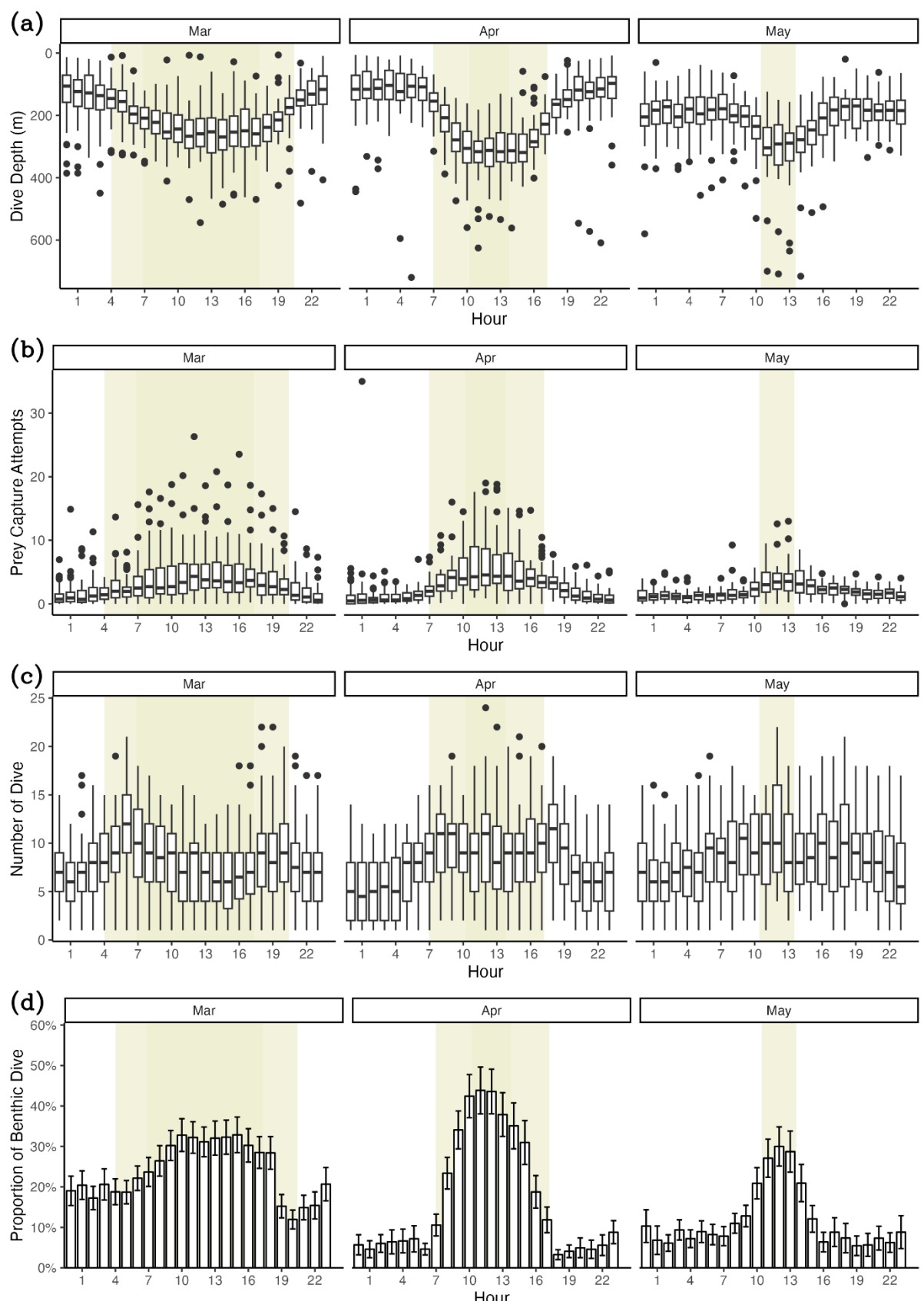

**Figure6. Diel variation in diving behaviors.** (a) dive depth, (b) prey capture attempts, (c) number of dives, (d) proportion of benthic dives. The yellow-shaded area denotes the duration of sunlight exposure during the day. The lighter yellow

shaded area indicates the period of daylight at the beginning of the month. In comparison, the darker yellow shaded area
represents the daylight period at the end of the month.

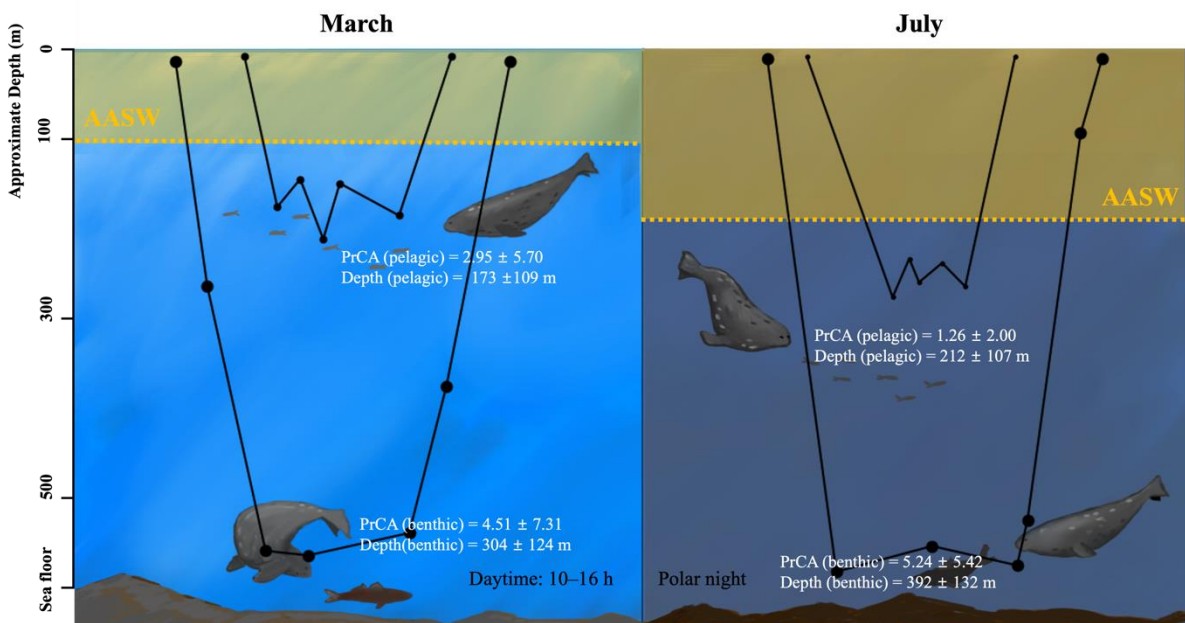

**Figure7. Schematic summary of seasonal variation in oceanographic conditions and foraging behaviors.** The area shaded in yellow represents the AASW, and the dashed line indicates the lower boundaries. In March, the AASW is positioned at shallower depths, whereas in July, the AASW shifts to deeper locations. AASW is a water mass less preferred by Weddell seals, possibly due to reduced prey availability, which appears to result in deeper dive depths during pelagic dives for Weddell seals. The black line graph in the figure represents typical examples of benthic and pelagic dives in March and July. The size of the dots is proportional to the PrCA values. PrCA and Depth values are presented as mean ± standard deviation. Note: The high variability of PrCA events results in SD values being larger than the mean values.


**Table 1. The best model for dive depth** AIC, BIC, and backward elimination approaches revealed that sex, season (month), and year are important variables for predicting dive depth.

| Dive Depth ~ Sex + Season + Year + (1\|IID) + corAR1(1\|IID) | | | |
|---|---|---|---|
| *Predictors* | *Estimates* | *CI* | *P* |
| (Intercept) | 203.79 | 181.65 – 225.93 | **<0.001** |
| Sex [Male] | -5.97 | -29.08 – 17.13 | 0.604 |
| Season [Apr] | -9.24 | -14.96 – -3.53 | **0.002** |
| Season [May] | 9.09 | 2.91 – 15.27 | **0.004** |
| Season [Jun] | 11.36 | 3.57 – 19.15 | **0.004** |
| Season [Jul] | 44.16 | 33.91 – 54.41 | **<0.001** |
| Year [2022] | 9.63 | -17.26 – 36.53 | 0.473 |
| Year [2023] | -9.85 | -38.74 – 19.04 | 0.495 |
| N IID | 44 | | |
| Observations | 59675 | | |

**Table 2. Post hoc (Tukey HSD) test for the "Season" variable included in the best model for dive depth**

| Group 1 | Group 2 | Mean Difference (Group 2 - Group 1) | Standard Error | Z value | Pr(>\|z\|) |
|---------|---------|-------------------------------------|----------------|---------|-----------|
| Mar | Apr | -9.245 | 2.916 | -3.17 | **0.0124** |
|  | May | 9.089 | 3.151 | 2.884 | **0.03** |
|  | Jun | 11.358 | 3.974 | 2.858 | **0.0323** |
|  | Jul | 44.16 | 5.23 | 8.443 | **<0.001** |
| Apr | May | 18.334 | 3.175 | 5.775 | **<0.001** |
|  | Jun | 20.603 | 4.006 | 5.143 | **<0.001** |
|  | Jul | 53.405 | 5.262 | 10.149 | **<0.001** |
| May | Jun | 2.268 | 4.024 | 0.564 | 0.9791 |
|  | Jul | 35.07 | 5.28 | 6.642 | **<0.001** |
| Jun | Jul | 32.802 | 5.661 | 5.794 | **<0.001** |


**Table 3. The best model for prey capture attempts** AIC, BIC, and backward elimination approaches revealed that water mass type, season (month), and dive type (benthic or pelagic) are important variables for predicting prey capture attempts.

**log(PCA_BTM + 1) ~ Water Mass + Dive Type + Season + (1|IID) + corAR1(1|IID)**

| Predictors | Estimates | CI | P |
|------------|-----------|-----|---|
| (Intercept) | 0.59 | $0.51 - 0.66$ | **<0.001** |
| Water Mass [HSSW] | 0.48 | $0.32 - 0.65$ | **<0.001** |
| Water Mass [ISW] | 0.31 | $0.27 - 0.35$ | **<0.001** |
| Water Mass [MSW] | 0.31 | $0.28 - 0.33$ | **<0.001** |
| Dive Type [Benthic] | 0.24 | $0.22 - 0.26$ | **<0.001** |
| Season [Apr] | 0.04 | $0.00 - 0.07$ | **0.044** |
| Season [May] | -0.11 | $-0.15 - -0.07$ | **<0.001** |
| Season [Jun] | -0.31 | $-0.36 - -0.26$ | **<0.001** |
| Season [Jul] | -0.18 | $-0.25 - -0.12$ | **<0.001** |
| N IID | 46 |  |  |
| Observations | 62317 |  |  |

**Table 4. Post hoc (Tukey HSD) test for the "Season" variable included in the best model for prey capture attempts**

| Group 1 | Group 2 | Mean Difference (Group 2 - Group 1) | Standard Error | Z value | Pr(>\|z\|) |
|---------|---------|-------------------------------------|----------------|---------|-----------|
| Mar | Apr | 0.03526 | 0.01752 | 2.012 | 0.24817 |
| | May | -0.10678 | 0.01951 | -5.473 | **< 0.001** |
| | Jun | -0.30808 | 0.02511 | -12.27 | **< 0.001** |
| | Jul | -0.182 | 0.03307 | -5.503 | **< 0.001** |
| Apr | May | -0.14204 | 0.01946 | -7.298 | **< 0.001** |
| | Jun | -0.34334 | 0.02517 | -13.639 | **< 0.001** |
| | Jul | -0.21725 | 0.03318 | -6.548 | **< 0.001** |
| May | Jun | -0.20131 | 0.02539 | -7.929 | **< 0.001** |
| | Jul | -0.07522 | 0.03333 | -2.257 | 0.15052 |
| Jun | Jul | 0.12609 | 0.03529 | 3.573 | **0.00295** |

**Table 5. Post hoc (Tukey HSD) test for the "Water Mass" variable included in the best model for prey capture attempts**

| Group 1 | Group 2 | Mean Difference (Group 2 - Group 1) | Standard Error | Z value | Pr(>\|z\|) |
|---------|---------|-------------------------------------|----------------|---------|-----------|
| AASW | HSSW | 0.48469 | 0.083488 | 5.806 | **<0.001** |
| | ISW | 0.308249 | 0.019305 | 15.967 | **<0.001** |
| | MSW | 0.305366 | 0.012401 | 24.625 | **<0.001** |
| HSSW | ISW | -0.176442 | 0.083729 | -2.107 | 0.124 |
| | MSW | -0.179325 | 0.082677 | -2.169 | 0.107 |
| ISW | MSW | -0.002883 | 0.015478 | -0.186 | 0.997 |

**Table 6. Regression analyses of dive parameters (dive depths, prey capture attempts per dive, number of dives per day) concerning presence of sunlight (day or night).**

**Dive Depth ~ Day/Night**

| Predictors | Estimates | CI | P |
|------------|-----------|-----|---|
| (Intercept) | 170.17 | 157.44 – 182.91 | **<0.001** |
| Day_boolTRUE | 76.4 | 73.25 – 79.55 | **<0.001** |
| N IID | 48 | | |
| Observations | 41733 | | |

**log(Prey Capture Attempts + 1) ~ Day/Night**

| Predictors | Estimates | CI | P |
| --- | --- | --- | --- |
| (Intercept) | 0.75 | 0.67 – 0.83 | **<0.001** |
| Day_boolTRUE | 0.32 | 0.30 – 0.35 | **<0.001** |
| N IID | 48 | | |
| Observations | 41733 | | |

**Number of Dives ~ Day/Night**

| Predictors | Estimates | CI | P |
| --- | --- | --- | --- |
| (Intercept) | 17.75 | 16.02 – 19.49 | **<0.001** |
| is_daytimeTRUE | -1.01 | -2.34 – 0.32 | 0.135 |
| N IID | 48 | | |
| Observations | 4614 | | |

**Proportion of Benthic Dives ~ Day/Night**

| Predictors | Estimates | CI | P |
| --- | --- | --- | --- |
| (Intercept) | -2.55 | -2.87 – -2.24 | **<0.001** |
| is_daytimeTRUE | 1.14 | 1.06 – 1.21 | **<0.001** |
| N $_{IID}$ | 48 | | |
| Observations | 41733 | | |

