# Peer review of "Seasonal foraging behavior of Weddell seals in relation to oceanographic environmental conditions in the Ross Sea, Antarctica"

_EGUsphere, 2023_

## Author Response (AR1)

**Dear Editorial Team of Biogeosciences**

Please find below our replies to the referee's comments, as requested. For your convenience, you will find the replies in blue in the corresponding lines below the respective comment. We thank you for considering our manuscript for publication in Biogeosciences. We particularly thank the Associate Editor, Andrew Thurber, for handling our manuscript with so much care.

**Authors Comments**

**Responses to comments by Reviewer #1 (Fabien Roquet)**

This manuscript describes the foraging behaviour of Weddell seals in the Ross Sea, using oceanographic and behavioural data logged using miniaturized loggers attached to the seals. While no ground-breaking result is obtained in this study, the data is of value and the analysis deserves its own publication conditional to major revisions.

I have in particular one major concern about the quality control of data. The authors cite in many places the publication of Yoon and Lee 2021. This publication is written in Korean in what appears to be a corean journal. This does not follow international standards and I am unable to follow what is written there. I believe the authors should treat this publication as a technical report, and assume the reader is unable to utilize it. For this reason, the current work should provide more extensive information about the different corrections that have been estimated and applied (Step 1 to Step 3 in section 2.2). The authors do not seem to be aware of the work of Siegelman et al 2019 either, which provides several recommandations for quality-control including density removal and thermal cell effect corrections.

⇒ We thank the reviewer for the valuable comments and for taking the time to review our work carefully. A point-by-point reply to all your comments can be found in blue. In addition, we have reinforced our research by adding 2023 seal data. Therefore, the revised manuscript included a total of three-year seal-CTD data.

We understand your major concern about data QC. However, seal-data were quality controlled following the international standards recommended mainly by Roquet et al. (2011) (Figure 2 of Yoon and Lee 2021). The figure below is based on Roquet et al. (2011) and

Siegelman et al. (2019). In this study, we applied the international standard QC method for low-resolution ascent profiles.

[Figure]

▲ Figure 2 of Yoon and Lee 2021
(Schematic procedures for quality control of two cases' CTD-SRDL data)

The only difference is HSSW (High Salinity Shelf Water) method as written in the original manuscript (L152–157). In the continental shelf region of the Ross Sea, the LCDW (Lower Circumpolar Deep Water) method (Roquet et al. 2011) could not be applied to seal data QC because LCDW is hardly found in this region. Instead of the LCDW, HSSW is a very stable feature in the deep layer of the western Ross Sea, including Terra Nova Bay. Therefore, we adjusted offsets of salinity and temperature data from each tag using HSSW properties observed from IBRV Araon survey during the austral summer of the corresponding year. Absolute values from ship-based CTD can be regarded as actual values because all CTD sensors of IBRV Araon were sent to SeaBird Electronics (SBE; Manufacturer) for sensor calibration one year before the observation period.

The salinity offset range for the 2021 seal data was from -0.16 to -0.03, and the temperature was not adjusted because the temperature of HSSW from the 2021 seal data was consistent with those from the ship-based CTD data. Temperature and salinity offsets for 2022 seal data were estimated as -0.03–0.23°C, and -0.38–-0.01, respectively. 2023 seal data were also quality controlled following the same method, and temperature and salinity offsets for 2023 seal data were estimated as -0.01–0.27°C, and -0.41–-0.01, respectively.

Moreover, Siegelman et al. (2019) suggested QC methods for thermal cell effect corrections, however, as you know, thermal mass correction could be only applied to high-resolution ascent and decent profiles. In this study, we only used low-resolution ascent profiles transmitted from CTD-SRDLs (profiles with 16 depths); thus, among several recommendations of Siegelman et al. (2019), we only applied the density removal algorithm (including Tag-by-tag data visualization in the figure above) regarding minimum $N^2$ ($N$ is the Brunt-Väisälä frequency) threshold as $1 \times 10^{-9} s^{-2}$. The figure below indicates vertical profiles of $N^2$ estimated from 2021, 2022, and 2023 QC completed seal data, respectively. All $N^2$ values are positive, which indicates that the density removal algorithm was successfully applied to the 2023 seal data.

[Figure]

▲ Vertical profiles of buoyancy frequency estimated from (a) 2021, (b) 2022, and (c) 2023 seal data

By the comment, we have added the information above on the seal data QC to the revised manuscript. In addition, we have added Supplementary Figure 1 by reproducing Figure 2 of Yoon and Lee 2021. Therefore, we referred to Supplementary Figure 1 instead of Yoon and Lee 2021 in the main manuscript. The figures below show a comparison of seawater properties between ship-based CTD data and QC-completed seal data for each year, and they captured well the characteristics of seawater in the western Ross Sea.

[Figure]

▲ (a) Comparison of T-S diagram between ship-based CTD data (blue) and QC-completed seal data for each year (gray; 2021, 2022, 2023) (b) Comparison of vertical profiles between ship-based CTD data during 2014-2022 (black) and QC completed seal data of 2021 (blue) (c) The same as (b), but for QC completed seal data of 2022 (green) (d) for 2023 (brown)

[Figure]

▲ Locations of ship-based CTD observations from 2014 to 2019 (circle) and ship-based CTD observations from 2020 to 2022 (Colored triangles; blue (Dec. 2020), green (Mar. 2022), brown (Dec. 2022)) in Terra Nova Bay. CTD data observed from 2020 to 2023 were mainly used for seal data QC.

If our paper is published in Biogeosciences, we plan to provide our raw data in the repository of the Korea Polar Data Center to an international consortium (MEOP, Marine Mammals Exploring the Oceans Pole to Pole) for sharing our data with seal-CTD researchers. We hope that the data can contribute to international collaborative research.

The analysis of water masses appears very rough. Figure 3 is very hard to read, and it has some strange features such as the MSW being "stuck" at 300m for most of the period. It would be nice to show some profiles and/or sections to get a better sense of what you are trying to show. ⇒ When we rechecked the boundary depth of MSW, the strange feature ("stuck") in the lower boundary depth of MSW was due to low-resolution profiles, not data QC problems. The low-resolution profiles have only data at 16 depths, so they cannot detect variations of the lower

boundary depth of MSW in the range of several tens of meters. By the comment, we have modified Fig. 3 because it may be misleading to readers. We remove the variations of boundary depths in Fig. 3 and add the figure below showing the vertical-temporal variations of seawater properties.

[Figure]

▲ (a) Hovmöller diagram of potential temperature around Terra Nova Bay. Gray, black and white solid lines represent -1.3, -1.7, and -1.9 °C isotherms, respectively. (b) Hovmöller diagram of salinity around Terra Nova Bay. Gray, black, and white solid lines represent 27.4, 27.8, and 28 kg/m³ isopycnals ($\sigma_\theta$), respectively.

To create the Hovmöller diagram, we used seal-tagging profiles around Terra Nova Bay within the longitude range between 160 and 170˚E and latitude range between 76 and 74˚S. Salinity and temperature from 1 to 600 m depth between February 15 and July 15 in 2022 were calculated using kriging. These variables were considered in the two-dimensional space of depth and time. To account for the spatiotemporal anisotropy, we scaled the values between 0 and 1 based on their maximum and minimum values, and multiplied the time values by 50.

According to the Hovmöller diagram, stratification weakens and the density difference between surface water and shelf water diminishes as winter advances. This is consistent with the seasonal variations of water masses suggested in the Results and Discussion Section of the original manuscript (L295–310 & L454–462).

One wonder also how much of your results depend on the different spatial sampling between the two years. I suggest the authors refine their analysis of hydrographic data to produce more specific results.

⇒ According to the map below, more seal data were obtained in 2021 and 2022 near the shelf-break and the eastern part of continental shelf regions compared to 2023. Due to this difference in spatial sampling, mCDW was identified more clearly in 2021 and 2022 compared to 2023. By the comment, we have added this result to the revised manuscript and refined the analysis of hydrographic data. However, this study primarily focuses on identifying changes in the foraging behaviors of Weddell seals. Therefore, we do not expand the analysis to include hydrographic changes, as these are beyond the scope of this study and warrant a separate investigation for further details with ship-based CTD data.

[Figure]

▲ Dive locations of seals tagged at Terra Nova Bay in the Ross Sea (blue, yellow, and brown dots indicated seal ARGOS locations in 2021, 2022, and 2023, respectively). The abbreviations CB, MB, PB, RB, DB, DT, JT, GCT mean Crary Bank, Mawson Bank, Pannell

Bank, Ross Bank, Drygalski Trough, Joides Trough, Glomar Challenger Trough, respectively. The dashed line represents the shelf break (at depths of 1000 and 2000 m), while the dotted line represents bathymetry at 200 m intervals (200-800 m).

Figure 6 is amongst the less informative I have every seen. That the mixed layer is deeper in winter than in summer shouldn't come as a surprise for anyone even remotely interested in oceanography. This cannot reflect the main novelty of this work. The authors need to clarify what is the main novelty of this study.

⇒ By the comment, we have added PrCA counts, dive depth, and photic condition information (day length) to Figure 6 and included a dive that can serve as an example. This figure visualizes our results that seals change their foraging attempts and depth in relation to the factors we considered (water mass, daylight hours, and benthic/pelagic dives).

[Figure]

▲ **Figure 1. Schematic summary of seasonal variation in oceanographic conditions and foraging behaviors.** The area shaded in yellow represents the AASW, and the dashed line indicates the lower boundaries. In March, AASW is positioned at shallower depths, whereas in July, the AASW shifts to deeper locations. AASW is a water mass less preferred by Weddell seals, possibly due to reduced prey availability, which in turn appears to result in deeper dive depths during pelagic dives for Weddell seals. The black line graph in the figure represents typical examples of benthic and pelagic dives in March and July. The size of the

dots is proportional to the PrCA values. PrCA and Depth values are presented as mean ± standard deviation.

Minor comment:

l. 35: this sentence seems to imply you are talking about "climate" change, because that's what you are describing earlier on, and maybe also because you use the word "adapt". Yet, the changes you describe in the text are related to diurnal/seasonal variability only. The abstract needs to be clarified.

⇒ By the comment, we have changed the word "adapt" to "respond". Also, we clarified that this study is on seasonal and diurnal variability. Accordingly, abstract's first and second sentences was modified as follows: "Understanding the foraging behavior of marine animals in Antarctica is crucial for assessing their ecological significance and responses to environmental changes, such as seasonal changes in seawater or diurnal light hours.".

l. 109-110: this accuracy numbers seem overly optimistic. See Siegelman et al. 2019 for recent estimates of accuracies.

⇒ We have modified this sentence as follows: "According to the specifications of the sensors of CTD-SRDL, the accuracy of temperature, pressure, and conductivity are ±0.005 ℃, 2 bBar, and ±0.01 mS cm$^{-1}$ (SMRU Instrumentation, 2024). However, low-resolution vertical profiles used in this study have a relatively lower accuracy for temperature (±0.04 ℃) and salinity (±0.03 g kg$^{-1}$) (Siegelman et al., 2019)."

**Responses to comments by Reviewer #2**

This manuscript describes environmental attributes that Weddell seals appear to favor for foraging within the Ross Sea, using satellite-linked relay loggers and accelerometers to document prey capture attempts. I think the authors have looked at the animals' foraging ecology from multiple different aspects for a comprehensive view of activities. I hope to see this published after some issues are addressed. There were many areas (identified in below comments) where there was insufficient detail to thoroughly understand methodology and how the authors had performed data processing. Overall, it was also difficult to discern the novelty of this study relative to other work that is referenced throughout the text where Weddell seals were tagged over the winter in the Ross Sea to determine important water masses they associate with. I think one very cool thing about this study that could be emphasized quite a bit more, is that these authors actually have prey capture events to compare between daytime/nighttime, season, and water masses. Prey capture and foraging success if often implied in these foraging ecology studies; however, to my knowledge this has not actually been measured before in Weddell seals overwinter. I also think the results here could be better into broader context with other Weddell seal foraging studies that have been conducted.

⇒ We thank the reviewer for valuable comments and for taking the time to review our work carefully. Based on the general comments on our manuscript about our methodology and data processing, we reflect on all the comments. Firstly, we added more detailed information about our methods to clarify the issues the reviewer raised. We have made clear that: 1) our data were obtained from the ARGOS satellite only, 2) the sample sizes of male and female, and 3) we used an SMRU accelerometer rather than a separate accelerometer. Secondly, we have clarified our data processing about prey capture estimation, dive threshold, and benthic diving determination. We have provided 1) detailed information about the prey capture attempts processing, 2) the dive threshold (6 m depth), 3) the benthic diving determination when seal diving was deeper than the bathymetric values, and 4) vertical travel speeds exceeding 5.1 m s$^{-1}$ were excluded. Thirdly, we have provided information on how we validated the models. Fourthly, we have emphasized the novelty and implications of our study in the discussion section. We provided that a) this is the first time that we have measured the prey capture attempts of Weddell seals in winter season using accelerometers, and b) the head acceleration data allows us to correlate foraging activities with the recorded environmental conditions, providing a clearer understanding of how these animals interact with their habitats. Lastly, we

have reinforced our research by adding 2023 seal data. Therefore, the revised manuscript included a total of three-year seal-CTD data. (Please check our responses to Reviewer 1's comments on the data QC). A point-by-point reply to all your comments can be found below and in blue.

Line 84-85: Would rephrase as 'Weddell seals are the deepest diving phocid with the exception of the elephant seals' (both southern and northern elephant seals dive deeper than Weddells)
⇒ By the comment, we have rephrased the sentence as follows: "They are ranked as the deepest diving phocid species except the southern (*Mirounga leonine*) and northern elephant seals (*Mirounga angustirostris*). "

Line 110: Should make clear whether all records were transmitted via ARGOS satellite, or whether some of these instruments were recovered.
⇒ By the comment, we have clarified that all data were obtained from Argos satellites, and no devices were recovered. A new sentence was added as follows: "All data obtained from CTD-SRDLs were received via Argos satellites and no instruments were recovered ."

Line 110: Since sex is used as a cofactor in model building (as authors state later in the Methods section) the sample sizes of male to female should be put somewhere in this paragraph.
⇒ By the comment, here we have mentioned our sample size and the number of sex. Also, we have clarified that two individuals with no sex determination in the field were excluded from the model analysis. The detailed individual information is presented in Supplementary Table 1. It has been added as follows: "Among the 64 seals, 27 were identified as females, and 35 were males based on their morphological feature. Two were not clearly distinguished in the field; hence, these were excluded from the model analysis for comparing the sexes (see Supplementary Table 1)."

Line 116-118: It is unclear whether this is a SMRU accelerometer. Or, whether this was a separate accelerometer that was attached alongside the SMRU tag. In either case, it needs to also be made clear what make and model the accelerometer was.
⇒ It was a SMRU accelerometer. By the comment, we have clarified that the source of the acceleration data is the accelerometer within the SMRU tag as follows: "Prey capture attempts were estimated from the transmitted head acceleration data obtained from the accelerometer

embedded in the CTD tags (referred to as "accelerometer processing," as detailed in the SMRU Instrumentation manual 2023)."

Line 124: Which logger divided the dive into 3 segments? Are the authors back to talking about the SMRU tag? The rest of this paragraph is confusing and should be clarified. Did the SMRU tags really divide each dive into 3 segments, as these instruments typically provide 4 inflection points within each dive. It sounds like that is the case here and the authors then did further processing by interpolating X number of midpoints and then the authors divided the dive into 3 segments: descent, bottom, and ascent. The 'dive threshold' also needs to be defined: is it that only dives >X m were retained in the dataset?

⇒ This paragraph was to explain our procedures, "summarizing the information on prey capture attempts", not about "summarizing the dives". We assumed that it was not clearly presented. To make it clearer, we have added a sentence about our PrCA estimation as follows: "Due to bandwidth limitations, summarized information was transmitted by dividing dives into three phases (descent, bottom, and ascent) and indicating the phase in which PrCA occurred, instead of transmitting the exact time and depth."

Also, the dive threshold (6 m depth) was added when we mention broken-stick points as follows: "Each dive was fitted to 12 broken-stick points (i.e., the depth at the first point below the dive threshold (6 m), 10 internal points, and the final point before the dive threshold (6 m))."

Line 195: How were dives that exceeded the IBCSO bathymetry (seals diving deeper than the 'bottom') treated?

⇒ When seal diving was deeper than the bathymetric values, the dives were regarded as benthic dives. We think that it is mostly due to the uncertain bathymetric data of the IBSCO or the slight difference between the exact diving location and the interpolated diving location of our results. To make it clear in the text, we have added a sentence in this paragraph as follows: "When seal diving was deeper than the bathymetric values, the dives were regarded as benthic dives"

Lines 199-200: change '<' to '>' for dives durations > 5760 and dive depths > 906 being excluded.

⇒ Thank you for pointing out this error. We have made the correction as suggested. Specifically, we have changed '<' to '>' to correctly indicate that dive durations greater than 5760 seconds

and dive depths greater than 906 meters are excluded. The revised sentence is as follows: "Furthermore, dives with durations that were too short or long and depths that were too great (dive duration = 0 s, dive duration > 5760 s, dive depth > 906 m; Heerah et al., 2013), and those characterized by vertical travel speeds exceeding 5.1 m s$^{-1}$ were excluded (Davis et al., 2003)."

Line 200: Would clarify that this is 'vertical travel speeds exceeding 5.1' (unless authors have also put a filter on horizontal distance traveled)

⇒ By the comment, we have clarified that the filter applies to 'vertical travel speeds exceeding 5.1'. We have not applied a filter on the horizontal distance traveled. The revised sentence is as follows: "Furthermore, dives with durations that were too short or long and depths that were too great (dive duration = 0 s, dive duration > 5760 s, dive depth > 906 m; Heerah et al., 2013), and those characterized by vertical travel speeds exceeding 5.1 m s$^{-1}$ were excluded (Davis et al., 2003)."

Line 219: Were these models run with REML or ML?

⇒ In our study, the model was fitted using REML. To make it clear in the text, this information has been added to the manuscript as follows: "The models were estimated using restricted maximum likelihood. ".

Line 219: There is no statement of model validation (checking for homoscedasticity etc).

⇒ By the comment, we calculated the R-squared values and performed Monte Carlo Cross Validation (CV) with a 4:1 train-test split and 100 iterations to validate the models. The R-squared value for the model with PrCA as the response variable and Dive type, Water Mass, and Season as explanatory variables was 0.139, with a Monte Carlo CV mean of 0.165 and a standard deviation of 0.007. For the model with Dive depth as the response variable and Sex, Year, and Season as explanatory variables, the R-squared value was 0.076, with a Monte Carlo CV mean of 0.078 and a standard deviation of 0.005. The model with PrCA as the response variable and Time of Day (Day or Night) as the explanatory variable had an R-squared value of 0.119, with a Monte Carlo CV mean of 0.118 and a standard deviation of 0.005. For the model with Dive depth as the response variable and Time of Day (Day or Night) as the explanatory variable, the R-squared value was 0.206, with a Monte Carlo CV mean of 0.204 and a standard deviation of 0.006. The model with Number of Dives as the response variable and Time of Day (Day or Night) as the explanatory variable had an R-squared value of 0.065,

with a Monte Carlo CV mean of 0.055 and a standard deviation of 0.018. Finally, for the model with Dive Type as the response variable and Time of Day (Day or Night) as the explanatory variable, the R-squared value was 0.175, with a Monte Carlo CV mean of 0.173 and a standard deviation of 0.007. We have added sentences in the method section as follows: "To ensure the robustness of our models, we performed Monte Carlo Cross Validation (CV) with a 4:1 train-test split and 100 iterations for each model. This approach allowed us to assess the stability and generalizability of the models. The standard deviations of the R-squared values were all below 0.02, further confirming the consistency and reliability of our models.".

Line 240: This whole paragraph incorporates a lot of discussion points into the Results section. These sentences especially that reference other works would be more appropriate in the Discussion section

⇒ We have modified this sentence as follows: "Moreover, the presence of MCDW was more discernable in the seal-tagging profiles compared to the ship-based CTD data obtained from the TNB, despite its limited occurrence (only 125 depths of 13,058 profiles) (Figs. 1 and 2; Supplementary Fig. 3). This prominence arises because of seals diving into the Drygalski and Joides troughs near the continental shelf break region (Fig. 1). Among seal data, more profiles were obtained near the shelf break and the eastern part of continental shelf regions in 2021 and 2022 than those in 2023 (Fig. 1). Due to this difference in spatial sampling, MCDW was identified more clearly in 2021 and 2022 compared to 2023 (Fig. 2; Supplementary Fig. 3). Furthermore, the ISW observed across the continental shelf region of the Ross Sea demonstrates a wider salinity range than the ISW observed in the TNB (Fig. 2), consistent with previous studies (ex) Budillon et al., 2011). In 2021, 2022, and 2023, properties of HSSW were well detected (Fig. 2) and it was mainly observed in the western part of the continental shelf region of the Ross Sea where polynyas exist (Fig. 2; Supplementary Fig. 3)."

Line 257: Percentage of dives made in MSW ?
⇒ The percentage of dives made in MSW was 76.76%. By the comment, we have added this value in the sentence as follow: " Based on our water mass definition, Weddell Seals performed many dives (76.76%) and high frequent observations of PrCAs in MSW."

Figure 2. Could the points be color coded by water mass? It is difficult to interpret.

⇒ We have added the inset below to the Fig. 2a, which shows the approximate TS range of each water mass.

[Figure]

Figure 3. I like seeing the depths of the seals and the depths of the water masses together; and the x-axis being categorical (month) for the boxplots makes sense. It is unclear how to make sense of the depth of the water masses with this x-axis. Is 'Mar' equivalent to March 1 for the continuous variable plotted for water mass? This should be made more clear.

⇒ By the comment, we have clarified the representation of the x-axis in Figure 3.

[Figure]

▲ **Figure 2. Temporal variation in dive depths of Weddell seals for 2021, 2022 and 2023 with Hovmöller diagram of seawater properties around Terra Nova Bay in 2022.** (a) Hovmöller diagram of potential temperature around Terra Nova Bay. Gray, black, and white solid lines represent -1.3, -1.7, and -1.9 ℃ isotherms, respectively. (b) Hovmöller diagram of salinity around Terra Nova Bay. Gray, black, and white solid lines represent 27.4, 27.8, and 28 kg m$^{-3}$ isopycnals ($\sigma_\theta$), respectively. (c) White and grey boxes indicate diving behaviors in 2021 and 2022, respectively, showing a tendency for deeper dives as austral winter approaches.

Figure 6 seems very general without a lot of information given that the AASW deepening is well known. I wonder if this might be better portrayed if the dive record of one seal is overlain on top of the schematic to show dive depth profiles (&with prey capture attempts marked) across a few days in March relative to the AASW; and dive depth of that same seal for a few days in July relative to AASW to show a representative example of the seal avoiding AASW if it is less preferred. Otherwise, this figure could probably be omitted.

⇒ By the comment, we have added PrCA counts, dive depth, and photic condition information (day length) to Figure 6 and included a dive that can serve as an example.

Tables: It seems odd that some of the variables stated to have a large impact on behaviors had very high p-values in the models (for example Table 1. Sex has a p value of 0.23 and year had a P value of 0.893 – did it really improve model fit enough to stay in the best fit model?). This is generally considered to be one of the drawbacks of stepwise approaches to model selection; or it can result from differences in ML versus REML methods.

⇒ Thank you for your comments regarding the model selection process and the inclusion of variables with high p-values. We appreciate the opportunity to clarify our methodology and results.

In our study, the model was fit using Restricted Maximum Likelihood (REML). We compared the models using both AIC and BIC values:

•       **AIC Comparison**: When comparing models using the AIC value, the best model included the variables sex, year, and season.

•       **BIC Comparison**: When comparing models using the BIC value, the best model included the variable only season.

Given these findings, the AIC value indicated that the model including sex, year, and season provided the best fit, whereas the BIC value suggested a simpler model excluding sex and year.

Line 277-278: Instead of 'variations in' would clarify which direction these shifts in behavior went in daytime versus night (greater proportion benthic dives, depths etc).

I also thought the Results said there was no difference in number of dives (i.e., foraging frequencies)?

⇒ By the comment, we have clarified the direction of the shifts in behavior between daytime and nighttime in the revised manuscript as follows: "Finally, a diel diving pattern among the

seals was observed, with an increase in the proportion of benthic dives, foraging frequency, diving depths, and the number of dives during the day compared to night."

In general, there were two things missing (for me) from the Discussion section. First, I think the most novel aspect of this paper is – that while there have been other Weddell seal tagging studies within the Ross Sea also looking at water masses that the animals associate with --- to my knowledge, this has never been paired with the addition of the accelerometers for prey capture events. This validates a lot of the ecological theories that have always been applied given the assumption that the animals are foraging a lot more in certain areas. It also highlights that even with a similar number of dives during the nighttime, animals are capturing less prey even though the animals (& prey) are likely shallower in the water column and it should be potentially less costly for the animals. That's pretty interesting! I am also aware of studies using accelerometry to document prey capture events in Weddell seals in the summer, but I am not aware of any such studies in the winter. I think some summer studies could be referenced for comparison between the breeding season, summer, and winter. I think more emphasis could be put on how the prey capture attempts validates important aspects of daily and seasonal foraging ecology.

⇒ Thank you for your insightful review. Based on your feedback, we have added the following paragraph to the Discussion section after the first paragraph to emphasize the novelty of our methods and implications of our study:

"To best our knowledge, this is the first to measure the prey capture attempts of Weddell seals in winter season directly using head acceleration with CTD. Previous studies have estimated foraging behaviors from indirect information, including horizontal location, vertical swim speed, dive time, and dive depth rather than being directly measured (Nachtsheim et al., 2019; Kokubun et al., 2021; Goetz et al., 2023). While these proxies are indirect indices and should be interpreted cautiously, acceleration data is particularly beneficial as it can directly detect PrCA, providing a more accurate measure of foraging activity (Heerah et al., 2019; Allegue et al. 2023). By directly measuring 3D head acceleration with CTD, we could obtain more reliable data on the foraging activities of the seals. This allows us to correlate foraging activities with the recorded environmental conditions, providing a clearer understanding of how these animals interact with their habitats. We presume that the combination of CTD and acceleration data offers a comprehensive view of both the physical environment and the behavioral responses of the seals, leading to more accurate and insightful conclusions.."

The other is I think that this would benefit for some discussion comparing the findings from this study with others that have tagged Weddell seals in the Ross Sea (were findings the same? – implying consistency across longer timespans? Or were some aspects different?). And also beyond the Ross Sea to put into context.

⇒ By the comment, we have compared the previous study in the Ross Sea on Weddell seals (Goetz et al.'s study in 2023). Goetz et al.'s finding was inconsistent with ours in foraging estimation. Thus, we specifically addressed such differences and discussed possible reasons as follows: "A previous study on Weddell seals in the Ross Sea showed that seasonal changes in foraging behavior were observed, with dive depth increasing and foraging activity intensifying from summer to winter (Goetz et al., 2023). The seasonal increase of dive depth agreed with our findings; however, their foraging results showed the contrary of our results. Goetz et al. (2023) observed that foraging of seals was the highest in winter of 2010 to 2012. However, we do not have supporting evidence to explain the difference. This could be due to the different seasonal prey availability in the Ross Sea between the two studies. In the Ross Sea ecosystem, the diet composition of Weddell seals exhibits considerable interannual variability (Goetz et al., 2017). The sea ice extent and the food availability in the Ross Sea for top predators can vary annually (Ainley et al., 2020). Such variations in sea ice extent can possibly influence plankton blooms and the seasonal prey abundance for seals between the two studies (Arrigo et al., 2004; Lorrain et al., 2009). Otherwise, the differences could result from the different measurements to infer foraging efforts. The results in winter foraging could be overestimated because the previous study used an indirect estimation (a track-driven metric) for foraging. The track-driven metric estimation is based on the assumption that all behaviors involving movement within a certain radius are associated with area-restricted search (ARS). However, behaviors other than foraging or movement (e.g., sleeping, resting) could also be regarded as ARS. Goetz et al. excluded haul-out periods from the FPT analysis to address this possibility. Nevertheless, haul-out times significantly decrease in winter while dive times increase (Boehme et al., 2016). This increase in dives during winter might suggest that the dives could be for purposes other than foraging, such as resting or sleeping."

---

## Referee Report (RR1)

The authors clearly did a lot of work including an additional year of dive data into analyses and they have clarified most of the questions I had about methodology. My remaining comments center on interpretation of some of the findings that have now been included in the Discussion section, and some important caveats that should be made clear. The Discussion section's paragraph topics also seemed to 'jump around', and I think edits for flow and concision in the Discussion would improve the manuscript.

The use of the word 'foraging' in many places in the discussion when comparing multiple studies that used very different metrics as proxy, was at many times quite confusing. For example, the authors stated 'their [Goetz 2023] foraging results showed contrary to our foraging results'. Goetz 2023 and other Weddell seal studies have used area restricted search and first passage time to infer areas/times associated with greater foraging effort. Conversely, this study used prey capture attempts and it seemed that these did not increase in the winter months as ARS had in previous work. There are a ton of possibilities that could account for these differences, and so the method used should be made clear throughout these paragraphs rather than clumping them all together as 'foraging'.

Abstract
Would remove 'diurnal' from 'diurnal light hours'

Methods
Line 120: What was the dose of Zoletil actually administered to the animals?

Line 135: I believe this should be reworded to say that --- dive descents were defined as the start of the dive until the first inflection point that exceeded 75% of maximum dive depth (?) As written, it sounds as though the descent only includes the first inflection point and this would only very infrequently be >75% max depth.

Line 275: Should refer to Figure 3 for dive depth

Lines 282-285: This feels like an incomplete/hanging sentence. It looks like the authors mean that PrCA's occurred most frequently in benthic dives, even though a small proportion of dives were classified as benthic (?). It also seems like this point would be better made at the very end of the paragraph.

Line 286: The authors said in the previous sentence that the most PrCA's occurred in MSW but here it sounds like it is actually HSSW?

Lines 293-294: I think it terminology should be kept consistent with this referring to PrCA's (instead of going back and forth with calling it foraging activities --- this could be taken to mean prey capture attempts or also could presume that all benthic dives are made with the intent that the animals are trying to forage, etc. It's more open ended).

Lines 302-303: I think the authors mean that the seals made more prey capture attempts in these water masses? It is a bit ambiguous when it is referred to as 'preference in foraging habitat' since that usually means whether the seals were simply present in a given location.

Lines 312-315: These two sentences essentially say the same thing and could be combined.

Lines 360-361: But there was no evidence that the two species were doing this. In fact, both the seals and emperor penguins made very similar proportions of deep dives exceeding ~350 m. The inter-specific competition was believed to be primarily between penguins and juvenile Weddell seals (not the adults to a great extent)

Lines 363-364: Again it becomes ambiguous when referred to simply as foraging activity, and this makes the paragraph more confusing. The authors should say how this study defined foraging. By defining it right away, the length of this paragraph could be significantly reduced by starting with the explanation that different metrics were used to measure foraging effort between the two studies.

Paragraph @ lines ~365-380. This paragraph should be edited for concision. This paragraph is also missing two important points. One is that prey may simply not be as predictable or more difficult to visualize in the dark winter months, so animals may in fact still be performing dives with the intent of foraging but have fewer prey capture opportunities. It also seems from the figures (4c & 6) that the decrease in PrCA's in ~July was primarily driven by a decrease in the number of pelagic dives (there was not a decrease in PrCA's in benthic dives). This point should be made in this paragraph, and that the frequency of PrCA's made during benthic dives appeared consistent throughout the year. The data here suggest that the animals relied more on benthic foraging late winter and this shift was reflected in season changes in PrCA's.

The organization of the Discussion section also jumps around quite a bit and could be edited for better flow. For example, this paragraph is followed by another about water masses, but then back to comparisons of foraging effort with previous work in lines 414-432. This paragraph and the paragraph @ lines 414-432 would seem to go together.

The authors note that 'jerk' may not accurately portray what is happening during capture of larger prey that require more handling. But a very important overall caveat that should also be included somewhere in this paper (perhaps in this paragraph) that what the authors have are prey capture ATTEMPTS. There is no way of knowing from these data whether the prey capture attempts were actually successful with the animal obtaining prey --- or not. There has also been some previous work (Fuiman, Davis, Williams) suggesting that prey capture attempts are more successful during daylight.

---

## Author Response (AR2)

**Dear Editorial Team of Biogeosciences**

**Authors Comments**
**Responses to comments by Reviewer**

The authors clearly did a lot of work including an additional year of dive data into analyses and they have clarified most of the questions I had about methodology. My remaining comments center on interpretation of some of the findings that have now been included in the Discussion section, and some important caveats that should be made clear. The Discussion section's paragraph topics also seemed to 'jump around', and I think edits for flow and concision in the Discussion would improve the manuscript.

⇒ We thank to the reviewer for the valuable comments and suggestions. As recommended, we have added descriptions regarding the method to determine the dosage of anesthetic administered to the Weddell seals, as well as the actual dosages used. In the Discussion section, we have toned down the potential interspecific competition between Weddell seals and emperor penguins and provided further explanation. We have also included a statement clarifying that our head measurement represents prey capture attempts, but may not reflect actual foraging success or the quality or quantity of prey obtained. Lastly, to make the Discussion paragraphs flow as presented in the Results section, we edited the paragraph topics by combining the separated comparisons with previous studies into a single paragraph. The detailed changes are described in our point-by-point responses.

The use of the word 'foraging' in many places in the discussion when comparing multiple studies that used very different metrics as proxy, was at many times quite confusing. For example, the authors stated 'their [Goetz 2023] foraging results showed contrary to our foraging results'. Goetz 2023 and other Weddell seal studies have used area restricted search and first passage time to infer areas/times associated with greater foraging effort. Conversely, this study used prey capture attempts and it seemed that these did not increase in the winter months as ARS had in previous work. There are a ton of possibilities that could account for these differences, and so the method used should be made clear throughout these paragraphs rather than clumping them all together as 'foraging'.

⇒ We agree that ARS is a different proxy since it is a prey search effort. In accordance with the comment, we differentiated the previous ARS estimation by Goetz et al.'s work (2023) with our head acceleration-based measurements. Since we measured the foraging behavior of Weddell seals by recording the number of head movement using an acceleration sensor, our method was to estimate the prey capture attempts. While we have used the term 'foraging' throughout the manuscript, we agree that it should be used more cautiously in certain sections, particularly when comparing our study with others, to avoid confusion. Also, we excluded the criticism on the first passage time estimation since this method is not exactly comparable to our study. Consequently, we have revised the Results and Discussion sections to use more specific terms such as 'prey search effort' to indicate the previous study by Goetz et al. (2023) and have added descriptions of the characteristics and limitations of these foraging proxies.

In Results section, we have revised as follows "Additionally, seals demonstrated higher PrCA events during the daytime, with an average of 4.89 foraging attempts per dive, compared to 2.13 attempts during the nighttime (Fig. 6b; Table 6).".

In Discussion section, we have revised as follows "A previous study on Weddell seals in the Ross Sea showed that seasonal changes for foraging effort were observed, with dive depth and prey search effort (estimated by search effort time in a given space) increasing from summer to winter (Goetz et al., 2023). Additionally, prey search effort was also higher in benthic dives (Goetz et al., 2023). The seasonal increase of dive depth agreed with our findings, but the prey search effort showed the contrary of PrCAs, our foraging measurement. Goetz et al. (2023) observed that the prey search effort was the highest in winter of 2010 to 2012. This could be due to the different seasonal prey availability across the seasons. During winter, our study indicated that prey capture was lower while the previous study by Goetz et al. (2023) showed the prey search effort was higher. To combine the two studies, it seems that the seals had to spend more time to search for prey despite the low foraging success in winter. Still, it is difficult to compare the two studies since there is an approximate 10-year difference. The diet composition of Weddell seals exhibits considerable interannual variability in the Ross Sea area (Goetz et al., 2017). The sea ice extent and the food availability for top predators can vary annually (Ainley et al., 2020). Such variations in sea ice extent can possibly influence plankton blooms and the seasonal prey abundance for seals between the two studies (Arrigo et al., 2004; Lorrain et al., 2009). Our measurement also has a limitation to compare the seasonal change. Prey capture attempts were estimated by 'jerk'

from acceleration sensors attached on heads. Prey capture attempts do not necessarily correlate with the quality or quantity of the prey successfully obtained. For example, jerk could be overestimated when handling larger prey items, as the number of handling movements increases (a case of Australian fur seals, *Arctocephalus pusillus doriferus*, Volpov et al., 2015). These limitations of foraging proxies may account for the observed differences in their seasonal trends.".

Abstract

Would remove 'diurnal' from 'diurnal light hours'

⇒ In accordance with the comment, we have made the correction as suggested. Rephrased sentence is as follow: "Understanding the foraging behavior of marine animals in Antarctica is crucial for assessing their ecological significance and responses to environmental changes, such as seasonal changes in seawater or light hours.".

Methods

Line 120: What was the dose of Zoletil actually administered to the animals?

⇒ Before anesthetization, the body length was roughly estimated by field researchers and an appropriate dosage of anesthetic was calculated and administered using the proportional relationship between the body length and weight of the Weddell seals. For example, a 2.2 m Weddell seal typically has an average weight of 310 kg, and a dosage of 4.8-6 ml of anesthetic is considered appropriate. We have added this information to the second paragraph of Method 2.1 as follows: "Before anesthetization, the body size was roughly estimated by the field researchers. Then, an appropriate dosage of anesthetic was administered using the proportional relationship between the body length and mass of the Weddell seals (Noren et al., 2008). The dosage administered to each individual (2 to 5 ml) is included in Supplementary Table 1."

Line 135: I believe this should be reworded to say that --- dive descents were defined as the start of the dive until the first inflection point that exceeded 75% of maximum dive depth (?) As written, it sounds as though the descent only includes the first inflection point and this would only very infrequently be >75% max depth.

⇒ We have made changes to clarify the sentences you were concerned about. Rephrased sentences are "Dive descents were defined as the start of the dive until the first internal point

that exceeded 75% of maximum dive depth. Similarly, the ascent phase began at the first internal point, where depths exceeded 75% of the maximum dive depth, and ending after the dive.".

Line 275: Should refer to Figure 3 for dive depth

⇒ Thank you for pointing out this error. We have made the correction as suggested. Rephrased sentence is "Figure 3 illustrates the seasonal changes in dive depth.".

Lines 282-285: This feels like an incomplete/hanging sentence. It looks like the authors mean that PrCA's occurred most frequently in benthic dives, even though a small proportion of dives were classified as benthic (?). It also seems like this point would be better made at the very end of the paragraph.

⇒ We have moved the mention of that part to the end of the paragraph to improve the flow and clarity as suggested. The paragraph has been revised as follows.

"In all three years, the Weddell seals tagged in this study exhibited distinct diving behaviors across months. Figure 3 illustrates the seasonal changes in dive depth. The dive depth shows an increasing trend from March to July, whereas the number of PrCA events decreases in June and July compared to March and April. When considering diving depth (p < 0.001; log likelihood ratio test between the best model and a model excluding the variable "season"), the shallowest dives were undertaken in April, whereas the deepest diving occurred in July (200 ± 137 m in April, 265 ± 154 m in July; mean ± standard deviation) (Fig. 3; Tables 1 and 2). In terms of PrCA (p < 0.001; log likelihood ratio test between the best model and a model excluding the variable "season"), the highest number was observed in April, whereas the lowest occurred in June (3.29 ± 6.11 in April, 1.56 ± 2.59 in June) (Fig. 4a; Tables 3 and 4). Additionally, PrCA values varied based on water mass and dive type (benthic or pelagic) (p < 0.001 for both; log likelihood ratio test between the best model and a model excluding the variables "water mass" and "dive type"). Based on our water mass definition, Weddell Seals performed many dives (76.76% of total dives) and high frequent observations of PrCAs (86.7% of total PrCA events) in MSW. The kernel density plots of dive distributions on a TS diagram are shown in Supplementary Fig. 4. Notably, Weddell seals displayed a higher number of PrCA events per dive in HSSW, MSW, and ISW compared to AASW (additional 1.14, 0.66, and 0.65 in PrCA per dive for HSSW, MSW and ISW, respectively; Tables 3 and 5). Our seals had 0.58 more PrCA during benthic dives than during pelagic dives (Fig. 4b;

Table 2), despite the fact that benthic dives were not predominant (11,741 out of a total of 64,014 dives, Fig. 4c). From March to July, PrCA consistently decreased during pelagic dives, whereas no significant decrease was observed during benthic dives (Figure 5, Supplementary Table 5).

Line 286: The authors said in the previous sentence that the most PrCA's occurred in MSW but here it sounds like it is actually HSSW?

⇒ The previous sentence means that the majority of total PrCA events occurred in MSW, while the current sentence indicates that the number of PrCAs per dive was higher in HSSW, MSW, and ISW. We have revised these sentences to clarify this distinction. Rephrased sentences are "Based on our water mass definition, Weddell Seals performed many dives (76.76% of total dives) and high frequent observations of PrCAs (86.7% of total PrCA events) in MSW. The kernel density plots of dive distributions on a TS diagram are shown in Supplementary Fig. 4. Notably, Weddell seals displayed a higher number of PrCA events per dive in HSSW, MSW, and ISW compared to AASW (additional 1.14, 0.66, and 0.65 in PrCA per dive for HSSW, MSW and ISW, respectively; Tables 3 and 5)."

Lines 293-294: I think it terminology should be kept consistent with this referring to PrCA's (instead of going back and forth with calling it foraging activities --- this could be taken to mean prey capture attempts or also could presume that all benthic dives are made with the intent that the animals are trying to forage, etc. It's more open ended).

⇒ We maintained the use of 'PrCA' instead of 'foraging' for greater accuracy. The revised sentence is as follows. "Additionally, seals demonstrated higher PrCA events during the daytime, with an average of 4.89 foraging attempts per dive, compared to 2.13 attempts during the nighttime (Fig. 6b; Table 6)."

Lines 302-303: I think the authors mean that the seals made more prey capture attempts in these water masses? It is a bit ambiguous when it is referred to as 'preference in foraging habitat' since that usually means whether the seals were simply present in a given location

⇒ We have revised the sentence to clarify that we are referring to the number of prey capture attempts. The revised sentence is "Among these, Weddell seals exhibited significantly higher number of PrCA events per dive for HSSW, MSW and ISW over AASW."

Lines 312-315: These two sentences essentially say the same thing and could be combined.

⇒ To make the text more concise, we have revised it into a single sentence as suggested. Revised sentence is "While these proxies are indirect indices and should be interpreted cautiously, acceleration data like the data our CTD obtained is particularly beneficial as it can directly detect PrCA, providing a more accurate measure of foraging activity (Heerah et al., 2019; Allegue et al., 2023).".

Lines 360-361: But there was no evidence that the two species were doing this. In fact, both the seals and emperor penguins made very similar proportions of deep dives exceeding ~350 m. The inter-specific competition was believed to be primarily between penguins and juvenile Weddell seals (not the adults to a great extent)

⇒ In our data, Weddell seals showed that approximately 13% of their dives exceeded 350 meters, with this percentage rising to 22% in July. Additionally, due to the lack of direct evidence, we have toned down the statement to only suggest the possibility. The revised sentences are "Additionally, seasonal variations in interspecific competition, particularly involving emperor penguins, another apex predator species in the Ross Sea year-round (Burns and Kooyman, 2001; Smith et al., 2012), could affect the foraging behavior of Weddell seals. In winter, emperor penguins must actively seek sustenance to nurture their offspring, potentially intensifying interspecific competition with Weddell seals (Burns and Kooyman, 2001). Given that the diving capacity of emperor penguins is lower than that of adult Weddell seals (Kooyman et al., 1980; Kooyman and Kooyman, 1995; Burns, 1999), Weddell seals may forage at greater depths to minimize interspecific competition. Notably, in our data, deep dives (exceeding 350 m) occurred at a rate of 22.4% in July for Weddell seals, whereas emperor penguins performed deep dives at a rate of less than 10% (Burns and Kooyman, 2001). This suggests a potential seasonal adjustment in foraging strategy, although direct evidence for this behavior remains limited.".

Lines 363-364: Again it becomes ambiguous when referred to simply as foraging activity, and this makes the paragraph more confusing. The authors should say how this study defined foraging. By defining it right away, the length of this paragraph could be significantly reduced by starting with the explanation that different metrics were used to measure foraging effort between the two studies.

⇒ To avoid confusion and aid understanding, we replaced the term 'foraging' in the mentioned sentence with the more specific term 'prey search effort.' The revised sentence is as follows: "A previous study on Weddell seals in the Ross Sea showed that seasonal changes for foraging effort were observed, with dive depth and prey search effort (estimated by search effort time in a given space) increasing from summer to winter (Goetz et al., 2023).".

Paragraph @ lines ~365-380. This paragraph should be edited for concision. This paragraph is also missing two important points. One is that prey may simply not be as predictable or more difficult to visualize in the dark winter months, so animals may in fact still be performing dives with the intent of foraging but have fewer prey capture opportunities. It also seems from the figures (4c & 6) that the decrease in PrCA's in ~July was primarily driven by a decrease in the number of pelagic dives (there was not a decrease in PrCA's in benthic dives). This point should be made in this paragraph, and that the frequency of PrCA's made during benthic dives appeared consistent throughout the year. The data here suggest that the animals relied more on benthic foraging late winter and this shift was reflected in season changes in PrCA's.

⇒ Thank you for your valuable comments and suggestions. According to our data, Weddell seals in the Ross Sea exhibited similar patterns in both PrCA per dive and daily PrCA, with high values in March and April and low values in June and July. We did not include this in the main text to avoid confusion due to an excess of results. Additionally, we have added another paragraph (3rd paragraph in Discussion section) to the discussion, addressing the observation that seals seem to rely more on benthic dives in winter. Added discussions is as follows: "Our data also showed that during the polar night in June and July, the PrCA per dive decreased in pelagic dives, while benthic dives showed no notable change (Figure 5, Supplementary Table 5). This suggests that benthic dives may play a crucial role in Weddell seals' foraging strategy during the winter months, when light conditions are diminished, making benthic prey potentially more reliable than pelagic prey.".

The organization of the Discussion section also jumps around quite a bit and could be edited for better flow. For example, this paragraph is followed by another about water masses, but

then back to comparisons of foraging effort with previous work in lines 414-432. This paragraph and the paragraph @ lines 414-432 would seem to go together.

⇒ As suggested, we have combined the comparison with previous studies into a single paragraph. The revised paragraph is as follows: "A previous study on Weddell seals in the Ross Sea showed that seasonal changes for foraging effort were observed, with dive depth and prey search effort (estimated by search effort time in a given space) increasing from summer to winter (Goetz et al., 2023). Additionally, prey search effort was also higher in benthic dives (Goetz et al., 2023). The seasonal increase of dive depth agreed with our findings, but the prey search effort showed the contrary of PrCAs, our foraging measurement. Goetz et al. (2023) observed that the prey search effort was the highest in winter of 2010 to 2012. This could be due to the different seasonal prey availability across the seasons. During winter, our study indicated that prey capture was lower while the previous study by Goetz et al. (2023) showed the prey search effort was higher. To combine the two studies, it seems that the seals had to spend more time to search for prey despite the low foraging success in winter. Still, it is difficult to compare the two studies since there is an approximate 10-year difference. The diet composition of Weddell seals exhibits considerable interannual variability in the Ross Sea area (Goetz et al., 2017). The sea ice extent and the food availability for top predators can vary annually (Ainley et al., 2020). Such variations in sea ice extent can possibly influence plankton blooms and the seasonal prey abundance for seals between the two studies (Arrigo et al., 2004; Lorrain et al., 2009). Our measurement also has a limitation to compare the seasonal change. Prey capture attempts were estimated by 'jerk' from acceleration sensors attached on heads. Prey capture attempts do not necessarily correlate with the quality or quantity of the prey successfully obtained. For example, jerk could be overestimated when handling larger prey items, as the number of handling movements increases (a case of Australian fur seals, *Arctocephalus pusillus doriferus*, Volpov et al., 2015). These limitations of foraging proxies may account for the observed differences in their seasonal trends.".

The authors note that 'jerk' may not accurately portray what is happening during capture of larger prey that require more handling. But a very important overall caveat that should also be included somewhere in this paper (perhaps in this paragraph) that what the authors have are

prey capture ATTEMPTS. There is no way of knowing from these data whether the prey capture attempts were actually successful with the animal obtaining prey --- or not. There has also been some previous work (Fuiman, Davis, Williams) suggesting that prey capture attempts are more successful during daylight.

⇒ As suggested, jerk measures only prey capture attempts and does not reflect the quality or quantity of the prey captured. Therefore, this difference could explain the discrepancy between our findings and those of the previous study (Goetz 2023), which used a different foraging proxy. We have added discussion as follows "Prey capture attempts were estimated by 'jerk' from acceleration sensors attached on heads. Prey capture attempts do not necessarily correlate with the quality or quantity of the prey successfully obtained. For example, jerk could be overestimated when handling larger prey items, as the number of handling movements increases (a case of Australian fur seals, *Arctocephalus pusillus doriferus*, Volpov et al., 2015). These limitations of foraging proxies may account for the observed differences in their seasonal trends.".